# A DNA alteration and methylation co-detection method for clinical purpose

Jiyan Yu[1,2,4], Chunhe Yang [1,2,4], Xintao Zhu[1,2,4], Zhankun Wang[1,2], Boping Xu[1,2], Ye Cai[1,2], Jingbo Zhao[1,2], Ruijian Guo[1,2], Wuzhou Yuan[1,2], Jianqing Wang[1,2], Bohao Dong [1,2], Frank Ron Zheng [1,2,3✉] & Shuang Yang [1,2,3✉]

## Abstract

**Traditional approaches for capturing genomic alterations and DNA methylation require separate assays, complicating clinical workflows and limiting sample utilization, particularly with low-input materials like cell-free DNA. To address these challenges, we introduce a streamlined approach combining mutation and methylation profiling via mutation-protective strand synthesis with modified deoxycytidine triphosphates, demonstrating high concordance with standard enzymatic methyl-seq and DNA-seq in both whole-genome sequencing of cell lines and targeted sequencing of clinical samples. In potential clinical contexts, incorporating multi-omics information with this approach modestly improve circulating tumor DNA (ctDNA) detection by ~12% in pre-treatment lung cancer patients ($N = 26$) while preserving specificity in healthy controls ($N = 13$), and reveal relationships between homologous recombination repair (HRR) gene function and homologous recombination deficiency (HRD) mediated by promoter methylation-driven biallelic loss of HRR genes in gynecologic cancer patients ($N = 27$). For practical convenience, this method was also implemented on qPCR platform with high performance (0.5% limit of detection). With its adaptability and potential utility in ctDNA detection and treatment, this approach holds promise for advancing clinical diagnostics.**

**Keywords** Mutation; Methylation; Co-detection; Minimal Residual Disease; Companion Diagnostics
**Subject Categories** Cancer; Chromatin, Transcription & Genomics; Methods & Resources

## Introduction

Genomic and epigenomic patterns within DNA molecules are pivotal for a broad spectrum of in-vitro diagnostic applications, including companion diagnostics (Fridlyand et al, 2013; Van Neste et al, 2011), early tumor detection (Cohen et al, 2018; Schrag et al, 2023), tissue of origin testing (Moran et al, 2016), and minimal residual disease (MRD) identification (Newman et al, 2014). Simultaneously capturing DNA alterations and methylation provides a more comprehensive view of biological processes and enhances diagnostic sensitivity. Notable examples include the development of a multi-target stool DNA test for colorectal cancer screening (Imperiale et al, 2014), and the integration of genomic and epigenomic assessments of circulating tumor DNA (ctDNA) in colorectal cancer patients (Parikh et al, 2021). In addition, combining mutation analysis with methylation profiling facilitates more precise therapeutic strategies, exemplified by the identification of *IDH1* mutations and *MGMT* promoter methylation in gliomas (Hegi et al, 2005; Mellinghoff et al, 2023). Furthermore, the interplay between genetic alterations and epigenetic modifications has been instrumental in understanding resistance mechanisms to targeted therapies. For instance, the use of poly ADP-ribose polymerase inhibitors (PARPi) is informed by combining mutations with methylation analysis of homologous recombination repair (HRR) pathway genes (Swisher et al, 2021), and the prediction of endocrine resistance in breast cancer is enhanced by integrating *PIK3CA* mutations with methylation of *ESR1* promoter regions (Mastoraki et al, 2018; Rasti et al, 2022). Despite these advancements, routine clinical application of simultaneous analysis of DNA alterations and methylation remains limited.

The primary barrier to more widespread adoption is the absence of methodologies that can handily simultaneously capture DNA alterations and methylation from limited clinical samples. Typically, samples are required to be divided into two separate reaction systems to detect DNA alterations and methylation independently, a step necessary to avoid confounding genuine C-to-T genetic changes with the conversion of unmethylated cytosine introduced by bisulfite or enzymatic treatment (Frommer et al, 1992; Vaisvila et al, 2021), but this division significantly reduces sample utilization efficiency and complicates clinical procedures. To address these challenges, several approaches have been developed to achieve simultaneous detection of DNA alterations and methylation without splitting samples (Fullgrabe et al, 2023; Liu et al, 2019; Wang et al, 2023c; Yan et al, 2022). Unfortunately, these methods have not fully met expectations due to various limitations, including the need for additional deconvolution steps to resolve information from the peculiar constructs formed during the

[1]Amoy Diagnostics Co., Ltd, 39 Dingshan Road, Haicang District, Xiamen, China. [2]Shanghai Xiawei Medical Laboratory Co., Ltd, 138 Xinjun Ring Road, Minhang District, Shanghai, China. [3]AmoyDx (Singapore) Pte. Ltd, Science Park Drive, Ascent #01-08, Singapore, Singapore. [4]These authors contributed equally: Jiyan Yu, Chunhe Yang, Xintao Zhu. ✉E-mail: frank.zheng@amoydx.com; shuang.yang@amoydx.com

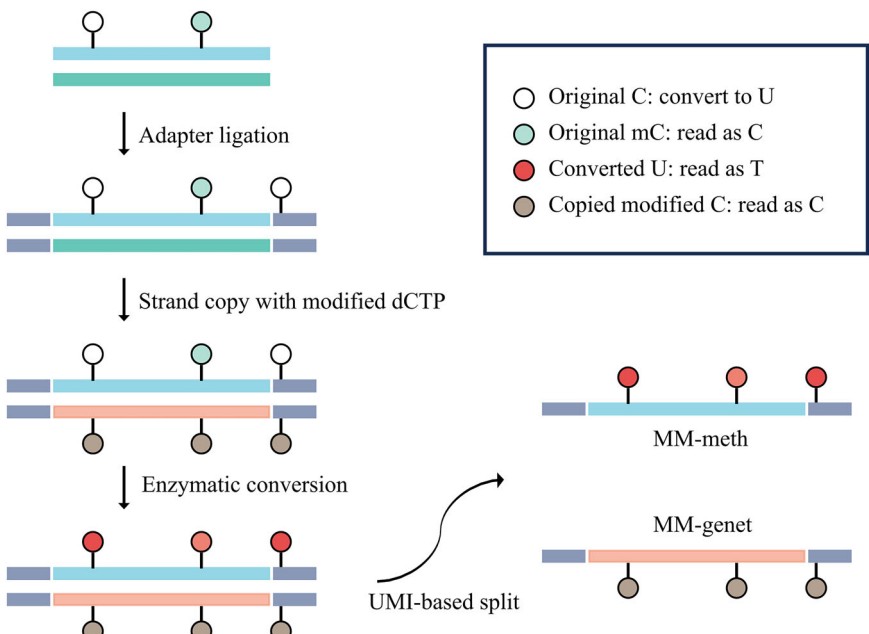

**Figure 1. Overview of MM-seq methodology.**

The MM-seq process involves several key steps, as illustrated in the figure. First, adapters are ligated to both ends of the DNA double strands. A copy strand is synthesized for both the Watson and Crick strands using modified dCTP, followed by a conversion process in which unmethylated cytosines are converted to uracils. The resulting library undergoes PCR amplification and sequencing. Produced paired-end reads are assigned to methylation analysis (MM-meth) if a cytosine is read as thymine in the UMI, or to mutation analysis (MM-genet) if the cytosine in the UMI remains unchanged.

process, which leads to data loss and limited platform adaptability, or the segregation of products mid-workflow to construct separate DNA mutation and methylation libraries, which remains cumbersome and less user-friendly for clinical use. Furthermore, most existing methods lack sufficient clinical validation to confirm their efficacy in diagnostics, with the exception of TET-Assisted Pyridine Borane Sequencing (TAPS), which still faces challenges in identifying mutations at methylated sites (Vavoulis et al, 2025).

In this study, we describe a versatile method for the co-detection of DNA mutation and methylation (MM), which employs modified dCTP to synthesize a mutation-protective strand, seamlessly integrating into conventional methylation library preparation (MM-seq) or can be utilized within a quantitative real-time PCR platform (MM-qPCR). When applied to clinical samples, this approach improves the sensitivity of ctDNA detection and offers diverse biomarkers, such as immune cell dynamics. In the context of companion diagnostics, the method provides detailed insights into gene function alterations, aiding in the interpretation of drug resistance and guiding therapeutic decision-making. With its adaptability across different platforms, this method shows promising potential for broad clinical applications.

## Results

### Principle of MM-seq

To develop a mutation and methylation co-detection method compatible with existing molecular detection techniques, it is

essential to preserve and distinguish genetic and epigenetic changes within a streamlined process. Our method introduces an additional step that generates a copy of the original DNA strand using modified dCTP, which resists deamination, before bisulfite/enzymatic conversion. This step conserves genetic modifications on the copy strand while retaining methylation changes on the original strand. Following the conversion step, the products can be subjected to various platforms, such as qPCR and next-generation sequencing (NGS), for downstream analysis. In high-throughput sequencing, duplex sequencing is used to separate genetic and epigenetic information. Specifically, cytosines in the original strand's unique molecular identifier (UMI) sequence undergo alteration after deamination, while those in the copy strand's UMI sequence remain unchanged, enabling the segregation of reads carrying genetic and epigenetic information.

To illustrate this co-detection workflow, we use MM-seq as an example (Fig. 1). Initially, adapters are ligated to both ends of the double-stranded DNA through a two-step ligation. The adapter sequence, excluding the UMI part, is modified by replacing unmodified cytosines with 5mC to ensure that the adapter sequence remains unchanged during deamination, thereby remaining compatible with standard library preparation and sequencing methods. Subsequently, complementary strand synthesis is performed for both Watson and Crick strands using modified dCTP to copy genetic alterations to the copy strand. During the ensuing enzymatic conversion, modified cytosines on the newly synthesized strands resist deamination, while unmodified cytosines on the original strand are deaminated to uracil. Consequently, the UMI sequence on the original strand undergoes C-to-T (G-to-A)

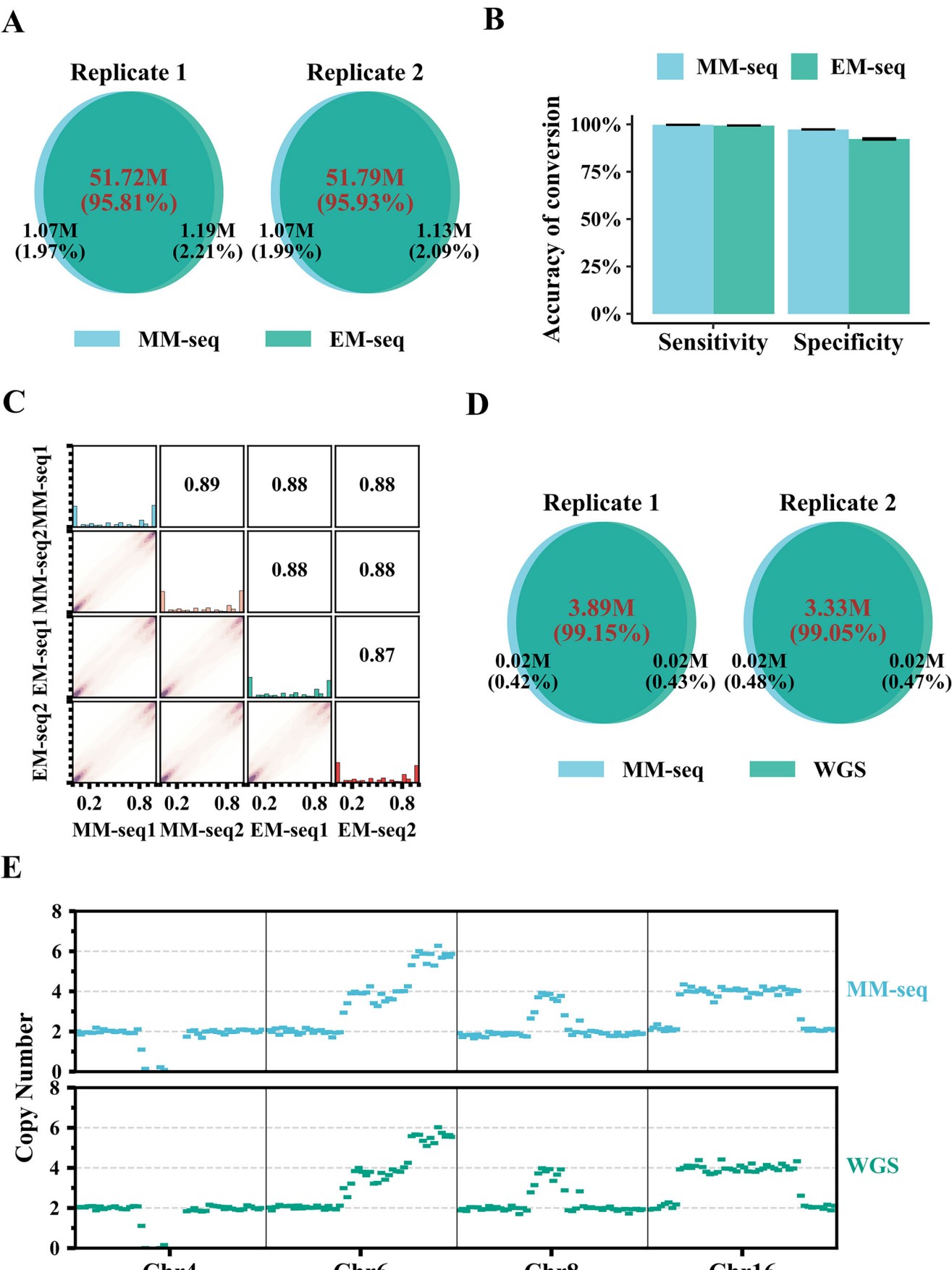

**Figure 2. Performance of MM-seq on genomic DNA.**

Two replicates from NA12878 cell lines were tested using MM-seq, WGS and EM-seq. (A) CpG site coverage by at least one read in EM-seq and MM-seq. Venn diagram illustrates the distribution of common shared and unique CpG sites in two replicates. (B) A bar plot shows the sensitivity and specificity of cytosine conversion in EM-seq and MM-seq. Sensitivity represents the proportion of converted unmethylated cytosines observed on fully unmethylated lambda DNA. Specificity showcases the proportion of unconverted methylated cytosines observed on fully methylated pUC19. (C) Correlation of CpG methylation among EM-seq and MM-seq. The density map exhibits the correlation of methylation levels of CpG sites with at least 5× coverage across different sequencing techniques in two replicates. (D) Venn diagram illustrating the number of shared SNPs identified from WGS and MM-seq in two replicates. (E) Concordance of four representative regions of CNV from a gold standard reference between MM-seq and WGS. Each bin represents a 10 kb genomic region.

conversion, while the UMI sequence on the copy strand remains unchanged. PCR amplification is then performed according to standard protocols, and the resulting constructs are subjected to high-throughput sequencing. The sequencing data include reads derived from the original strand, which carries methylation information (MM-meth), and reads derived from the synthesized strand using modified dCTP, which carries sequence information (MM-genet). Segregation of the two types of reads is achieved by identifying the differences in the UMI sequence. Specifically, reads whose UMI sequence perfectly matches the primary UMI sequence from the UMI pool are assigned to MM-genet, while reads whose UMI sequence matches the C-to-T (G-to-A) converted UMI sequence are assigned to MM-meth (Appendix Table S1). MM-genet and MM-meth data are subsequently used for the assessment of genomic alterations and methylation, respectively.

Given the reliance on a UMI-matching strategy for read assignment, the accuracy of this process depends heavily on the precision of the UMI sequence. Incomplete conversion of unmodified cytosines can introduce base errors in the original strand's UMI sequence, while deamination of modified cytosines can lead to base errors in the copy strand's UMI sequence. During enzymatic conversion, modified cytosines are oxidized by TET to protect against deamination to uracil (read as thymine) (Schutsky et al, 2017). The efficiency of this protection and resistance to deamination varies among different cytosine analogs, leading to varying levels of C-to-T (G-to-A) errors in reads, including the UMI sequence. To identify the most suitable modified dCTP for the protective strand synthesis step, MM-seq was implemented on three replicates of the NA12878 cell line, assessing four previously reported cytosine analogs: 5-methyl-dCTP (dmCTP), 5-hydroxymethyl-dCTP (dhmCTP), 5-carboxy-dCTP (dcaCTP), and 5-propynyl-dCTP (dpyCTP). Libraries using dpyCTP demonstrated the highest proportion of perfectly matched UMIs (95.45%, Appendix Fig. S1A) compared to those using dmCTP (83.76%), dhmCTP (92.21%), and dcaCTP (89.00%). This observation aligns with previous reports identifying dpyCTP as a deamination-resistant cytosine analog (Wang et al, 2023a). Consequently, dpyCTP also exhibited the lowest C-to-T (G-to-A) substitution rate across the exome (0.15% for dpyCTP; 0.58% for dhmCTP, 0.82% for dcaCTP, and 2.33% for dmCTP, Appendix Fig. S1B), closely matching the rate observed in standard WGS (0.03%). Additionally, under identical conditions of seven PCR cycles, libraries using dpyCTP yielded an average of 972 ng (Appendix Fig. S1C), equivalent to dmCTP (984 ng) and higher than dhmCTP (850 ng) and dcaCTP (679 ng), indicating more usable reads under the same experimental conditions. Therefore, dpyCTP was selected for synthesizing the protected strand in subsequent experiments.

## Evaluation of MM-seq by whole-genome sequencing of NA12878

As a proof of concept, we compared MM-seq with two established methodologies, EM-seq and WGS, commonly used for assessing methylation and genomic alterations. Whole-genome sequencing was performed in duplicate using 30 ng of genomic DNA (gDNA) from the NA12878 sample, incorporating two spike-in controls: fully unmethylated lambda DNA and fully methylated pUC19. For the two replicates sequenced with MM-seq, 550 million paired-end reads were generated, which were then subjected to UMI-based splitting, yielding 181 and 179 million read pairs for MM-meth, and 307 and 311 million read pairs for MM-genet.

To ensure comparability across different technologies, paired-end reads from EM-seq and MM-meth were downsampled to 165 million pairs for methylation analysis. The results showed that MM-seq had a comparable template utilization (11.36× mean deduplicated coverage) to EM-seq (11.88×), with similar mapping rates and sequencing duplicates (Appendix Fig. S2A). MM-seq, like EM-seq, demonstrated consistent coverage around CpG islands (CGIs) and their flanking regions, indicating uniform CGI coverage by MM-seq (Appendix Fig. S2B). Consequently, among 56 million CpG sites across the genome, MM-seq covered approximately the same amount with at least one read (93.61%) as EM-seq (93.77%) and shared over 95% of the covered CpGs (Fig. 2A). In evaluating methylation estimation (Fig. 2B), EM-seq demonstrated high sensitivity for converting unmethylated cytosines in lambda DNA (99.49%), consistent with previous reports (>99.4%) (Vaisvila et al, 2021). MM-seq achieved comparable conversion sensitivity (99.89%) and exhibited high specificity (97.43%), defined as the ratio of unconverted cytosines on the pUC19 contig. For CpGs covered by at least five reads, MM-seq showed a strong correlation with EM-seq in base-level CpG methylation across the genome (Pearson's $r > 0.88$, $P$ value < 0.01, Fig. 2C). Consequently, the methylation levels of CpG sites across the entire genome measured by MM-seq and EM-seq were comparable (50.80% and 49.17%, respectively), as were the cytosine methylation levels within CHG and CHH contexts (0.11% and 0.43%, respectively; Appendix Fig. S2C).

For the analysis of genetic alterations, we sampled an equal number of 300 million paired-end reads for WGS and MM-genet, processing them using the same pipeline. The data revealed that the mean deduplicated coverage in MM-genet (21.30×) slightly exceeded that in WGS (19.55×), likely due to a lower ratio of PCR duplicates in MM-genet (Appendix Fig. S3A). Although the C-to-T (G-to-A) substitution rates across the exome in MM-genet (0.07%) were marginally higher than in WGS (0.04%) (Appendix Fig. S3B), over 99% of detected single-nucleotide polymorphisms

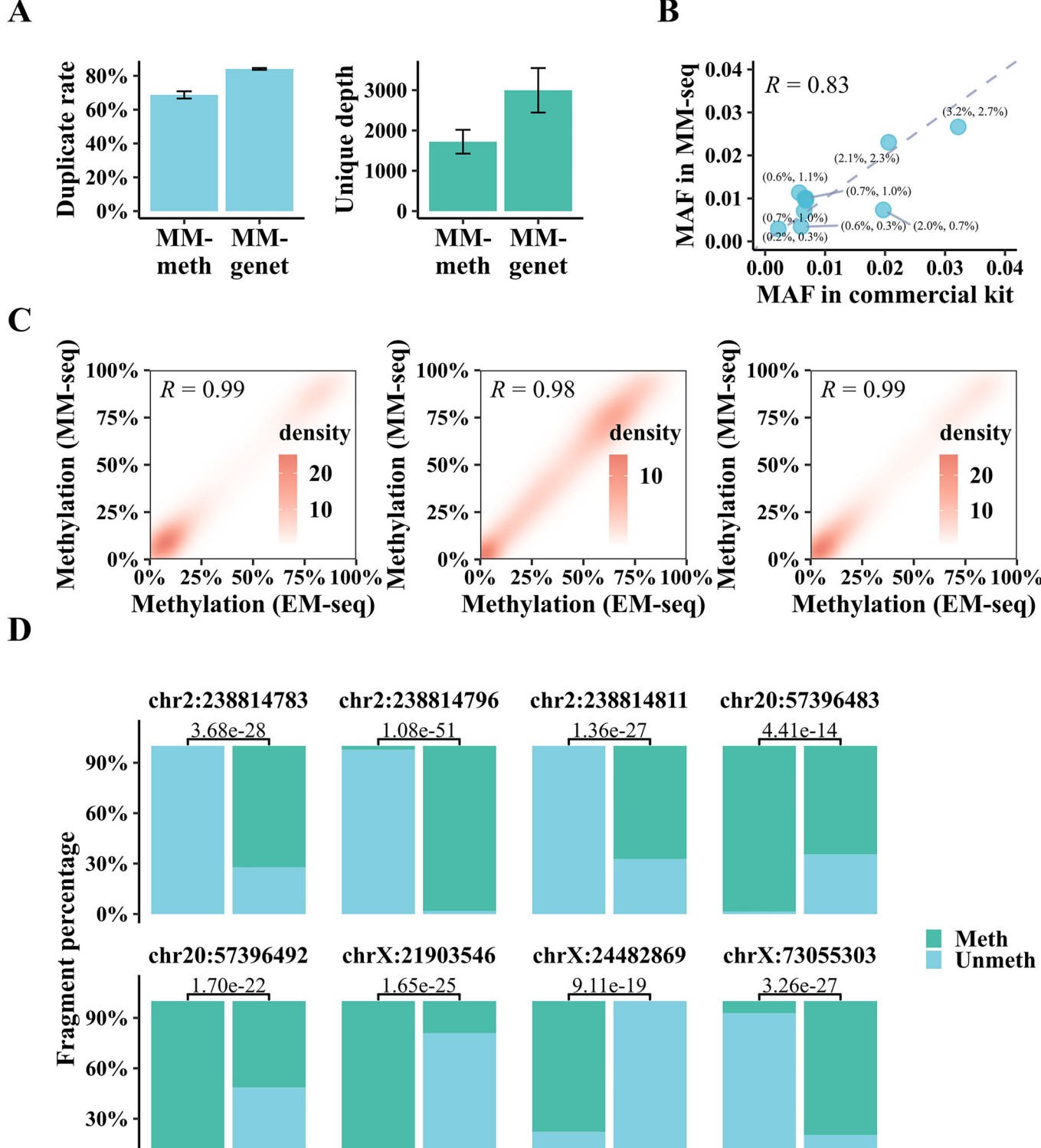

**Figure 3.  Assessment of MM-seq in clinical scenarios.**

(A) QC metrics of MM-seq combined with target enrichment. The ratio of PCR duplicates is shown on the left, and mean deduplicated coverage is shown on the right for MM-genet and MM-meth produced with 15 ng cfDNA from 13 healthy donors. Data are presented as mean ± standard deviation. (B) Concordance of MAF of somatic mutations identified in cfDNA samples from ten NSCLC patients using MM-seq and a commercial kit. MAF values are displayed for each mutation, with the former representing the commercial kit and the latter representing MM-seq. (C) Correlation of CpG methylation between MM-seq and EM-seq in cfDNA samples. Methylation profiles of three cfDNA samples were analyzed using MM-seq and EM-seq. The density plot represents CpGs at different methylation levels. (D) ASM measured by MM-seq. Bar plots show the percentage of fragments with methylated and unmethylated cytosines for each allele of SNP. Genomic coordinates represent CpG near five SNPs related to known ASM (Fisher's exact test for testing allele-specific significance).

(SNPs) were shared by MM-seq and WGS when considering genomic loci covered by at least ten reads (Fig. 2D), confirming the concordance between MM-genet and WGS in SNP detection. To further assess MM-seq's reliability in evaluating copy number variations (CNVs), we scrutinized the high-confidence CNVs documented for the NA12878 gold standard reference sample using both WGS and MM-seq (Haraksingh et al, 2017). The CNVs initially identified in the WGS dataset exhibited parallel occurrences within nearly identical genomic regions in MM-genet (Fig. 2E), demonstrating that MM-seq is capable of accurately characterizing CNVs.

## Assessment of applying MM-seq in target sequencing

In clinical practice, targeted sequencing is typically preferred because it focuses on specific regions of interest that are clinically relevant, thereby maximizing the utility of sequencing efforts and offering a more cost-effective approach. This method also provides higher coverage, which is crucial for achieving the required sensitivity and specificity, particularly in cell-free DNA (cfDNA) samples. MM-seq is theoretically compatible with target enrichment, allowing for the capture of regions using a probe pool designed to assess methylation, mutation, or a combination of both. Considering the common clinical scenario of limited cfDNA availability from blood draws, we initiated the study by preparing MM-seq libraries using 2 ng and 10 ng of cfDNA, as well as 80 ng of gDNA from healthy donors. After seven cycles of PCR amplification, MM-seq successfully produced library yields of 56 ng, 115 ng, and 1554 ng for the 2 ng, 10 ng, and 80 ng DNA inputs, respectively (Appendix Fig. S4A), demonstrating MM-seq's potential applicability across various sample types and low-input DNA scenarios. Further, MM-seq libraries were prepared with 15 ng of cfDNA, which represents the typical lower limit for DNA input in cancer genetic testing, from 13 healthy donors. To satisfy the need of detecting low-frequency somatic mutations from cfDNA samples, it is essential to obtain as many as possible unique reads for MM-genet. We employed a strategy that involved increasing the number of pre-amplification cycles to compensate for the template loss during conversion, and three cycles of copy strand synthesis were selected as the optimal condition for our experiments involving cfDNA samples (Appendix Fig. S4B). These libraries prepared with 15 ng of cfDNA were captured using a hybrid panel designed for ctDNA assessment, incorporating methylation probes targeting ~21 kb of the genome and sequence probes targeting ~38 kb (detailed in the methods section under ctDNA detection). MM-seq generated a mean of 2.1 million paired-end reads for MM-meth and 20.2 million paired-end reads for MM-genet, resulting in a mean on-target read duplicate rate of 68.71% (ranging from 64.19

to 71.56%) and a mean unique depth of 1722× (ranging from 1036× to 2115×) for MM-meth, and a mean on-target read duplicate rate of 84.08% (ranging from 83.00 to 85.37%) and a mean unique depth of 2997× (ranging from 1890× to 3976×) for MM-genet (Fig. 3A), sufficient for most clinical scenarios.

To evaluate MM-seq's ability to reliably identify rare mutations and methylation, we compared it with standard capture-based deep sequencing libraries. For mutation detection, 30 ng cfDNA samples from each of the ten non-small cell lung cancer (NSCLC) patients were used to prepare libraries using standard DNA-seq. The remaining cfDNA from each sample (ranging from 4.56 to 29.75 ng, Table EV1) was used to prepare MM-seq libraries, incorporating three cycles of copy strand synthesis to enhance mutation detection. Both MM-seq and standard DNA-seq libraries were captured using a commercial lung cancer panel (LC10, see gene list in Appendix Table S2). Ten NSCLC-related driver mutations were consistently observed in both MM-seq and DNA-seq libraries, with mutant allele frequencies (MAF) showing a high correlation (Pearson's $r = 0.83$, $P$ value < 0.01, Fig. 3B). For methylation assessment, 10 ng cfDNA samples from three cfDNA samples from healthy donors were used to construct MM-seq and EM-seq libraries, which were captured with methylation probes targeting ~178 kb of the genome. The results revealed high CpG methylation correlations between the two techniques (Pearson's $r > 0.98$, $P$ value < 0.01, Fig. 3C). In addition, pUC19 controls at various methylation levels (5%, 2%, 1%, 0.5%, 0.2%, 0%) were diluted using fully methylated and unmethylated pUC19 and added to six cfDNA samples to construct MM-seq libraries, which were then captured using probes designed for the pUC19 contig. The CpG methylation levels of pUC19 measured by MM-seq showed a strong correlation with theoretical dilution gradients (Pearson's $r = 1$, $P$ value < 0.01, Appendix Fig. S4C), and the beta value profile of single CpG sites across the pUC19 genome aligned with their theoretical concentrations (Appendix Fig. S4D).

Notably, duplex sequencing in MM-seq allows for the tracking of MM-genet and MM-meth reads derived from the same DNA molecules by identifying identical genome mapping coordinates and paired UMI sequences. This capability enables the phasing of genetic and epigenetic properties in cis, such as the detection of allele-specific methylation (ASM), by using hybrid probes to capture both the methylome and genome in the same genomic region. We selected ASM candidates previously reported in NA12878 (Yan et al, 2022), which included eight CpGs around five SNPs at the *RAMP1*, *GNAS*, *XIST*, *PAK3*, and *MBTPS2* loci. Methylation and sequence probes targeting these SNPs were designed to capture an MM-seq library prepared with 30 ng of gDNA from the NA12878 sample. By pairing reads from MM-genet and MM-meth based on mapping positions and UMI sequences,

36.28% of fragments in MM-meth found their pairs in MM-genet and were allocated to discrete alleles of the target SNPs. Significant differences in methylation between wild-type and mutant-type alleles were observed in all eight CpG candidates (*P* value < 0.001, Fig. 3D), demonstrating MM-seq's ability to resolve combinations of genetic and epigenetic information on the same DNA molecule.

## MM-seq supports ctDNA detection

Panel-based mutation detection poses substantial challenges in ctDNA detection, particularly in low tumor burden scenarios, where the recovery of mutation-bearing ctDNA fragments is limited (Deveson et al, 2021; Song et al, 2022). Recent research has highlighted the potential of ctDNA or immune-derived cfDNA methylation in tumor detection (Fox-Fisher et al, 2021; van der Pol and Mouliere, 2019; Wang et al, 2023b), and its integration with genomic alterations may improve tumor detection sensitivity (Gu et al, 2024; Parikh et al, 2021), while simultaneously providing valuable insights into cancer prognosis and therapeutic response (Dillinger et al, 2022; Oliver et al, 2022). To verify this idea through MM-seq, we designed a variant-detecting panel covering 27 genes (LC27, see full gene list in Appendix Table S2), which includes driver genes and those recurrently mutated in NSCLC, utilizing data from The Cancer Genome Atlas (TCGA) and the CHOICE cohort (Zhang et al, 2019). This panel was further expanded with eight NSCLC-specific methylation markers (Appendix Table S3). In addition, we incorporated 17 methylation markers known to identify immune-derived cfDNA based on cell type-specific methylation patterns, providing insights into remote immune processes that reflect the tumor dynamics (Fox-Fisher et al, 2021). These immune-specific methylation markers were validated in ten healthy donors, demonstrating a similar correlation trend as previously reported between methylation-inferred gDNA from various immune cell types and the flow cytometry (FCM) benchmark, as well as between methylation-inferred cfDNA and the FCM benchmark (Appendix Fig. S5A).

We collected 26 pre-treatment cfDNA samples from NSCLC patients and 13 cfDNA samples from healthy donors, applying two approaches for ctDNA detection using equal 20 ng cfDNA per method: conventional hybrid capture-based DNA-seq with the variant-detecting panel, and MM-seq, which employed the panel combined mutation and methylation information. Paired white blood cell (WBC) DNA from all samples showing somatic mutations was also tested with standard DNA-seq to confirm the somatic nature of detected mutations. Both methods achieved 100% specificity, with no healthy control samples identified as ctDNA-positive. MM-seq, by integrating mutation and methylation signals, increased the ctDNA-positive detection rate to 26.92% (7/26), compared to 15.38% (4/26) using the traditional mutation-based DNA-seq (Fig. 4A). In the four samples where DNA-seq identified ctDNA, the variant allele frequencies (VAF) of detected mutations were highly correlated with the ctDNA fractions inferred through methylation in MM-seq (Pearson's *r* = 1, *P* value < 0.01, Fig. 4B), indicating MM-seq may aid in estimating ctDNA burden using methylation data. Among the seven ctDNA-positive samples identified by MM-seq, two samples were detected by both methylation and mutation, and five samples were detected by methylation alone. Significantly, all four patients detected by DNA-seq were in advanced stages, whereas MM-seq not only detected four late-stage patients but also identified 3 early-stage patients (Fig. 4C; Appendix Table S4), indicating possible utility in

detecting low-burden tumor. Furthermore, in one instance where a *FAT1* mutation was detected with a VAF of 4.09% in DNA-seq and 3.59% in MM-seq, the methylation-inferred ctDNA fraction was only 0.04%. The mutation was excluded as it was also detected in the paired WBC sample, suggesting that methylation data can be instrumental in distinguishing true tumor-derived mutations from clonal haematopoiesis of indeterminate potential (CHIP) in cfDNA.

No significant differences in immune cell subsets except lymphocytes were observed between NSCLC patients and healthy controls (Appendix Fig. S5B), and NSCLC patients exhibited a significantly higher neutrophil-to-lymphocyte ratio (NLR) compared to healthy individuals (Student's *t* test *P* value < 0.05, Fig. 4D), consistent with prior findings that elevated NLR is commonly observed in cancer patients and is associated with poorer prognosis (Forget et al, 2017; Mosca et al, 2023). We did not include NLR as an indicator for ctDNA detection, as determining an appropriate threshold to differentiate cancer patients from healthy individuals is tricky in this context, but could be addressed with further investigation in larger cohorts.

## MM-seq aids in therapeutic guidance

Tumor resistance to targeted therapies often arises from secondary mutations or epigenetic modifications. For instance, resistance to PARPi may develop due to the loss of methylation in HRR genes, even when homologous recombination deficiency (HRD) caused by HRR gene dysfunction is present (Nesic et al, 2021; Swisher et al, 2021), necessitating an integrated analysis of both genetic and methylation factors. To investigate this, we analyzed 27 formalin-fixed paraffin-embedded (FFPE) samples from ovarian and breast cancer patients, all of whom had BRCA1 loss-of-heterozygosity (LOH). These samples were categorized into HRD (*N* = 16) and homologous recombination proficiency (HRP, *N* = 11) groups based on HRD scores (Table EV2) (Telli et al, 2016). Using data from TCGA, we identified expression-correlated CpGs in the promoter regions of eight HRR genes (Appendix Fig. S6A) and designed corresponding probes to form a 1.24 kb panel. This panel was combined with a 741 kb pan-cancer genomic panel to capture the prepared MM-seq library, using 30 ng of DNA from each sample, to evaluate promoter methylation in eight HRR genes and genetic alterations in 16 HRR genes. Our analysis identified LOH in *RAD51C*, *RAD51D*, *BRIP1*, and *CDK12* in all samples, as these genes are located near *BRCA1* on chromosome 17. Promoter methylation was observed in *BRCA2* and *FANCA* in all samples, *BRCA1* promoter methylation was detected in one-third of the samples, and *RAD51C* promoter methylation was observed in one sample. Only one pathogenic mutation was detected in *RAD51D*, and no homozygous deletions were identified in any of the samples (Fig. 5A).

Given the absence of biallelic loss through homozygous deletions or two concurrent deleterious mutations, we examined biallelic loss manifested as LOH combined with either promoter methylation or deleterious mutations (Rempel et al, 2022). A significant correlation was observed between biallelic loss and HRD (*P* value < 0.01). Specifically, biallelic loss of function in at least one HRR gene was identified in 75% (12/16) of HRD patients, compared to 9.09% (1/11) of HRP patients. Still, four HRD patients exhibited no biallelic loss in any HRR genes (Appendix Fig. S6B). Focusing on individual HRR genes, *BRCA1*, *RAD51C*, *CDK12*, and *BRIP1* exhibited LOH in all samples, so promoter

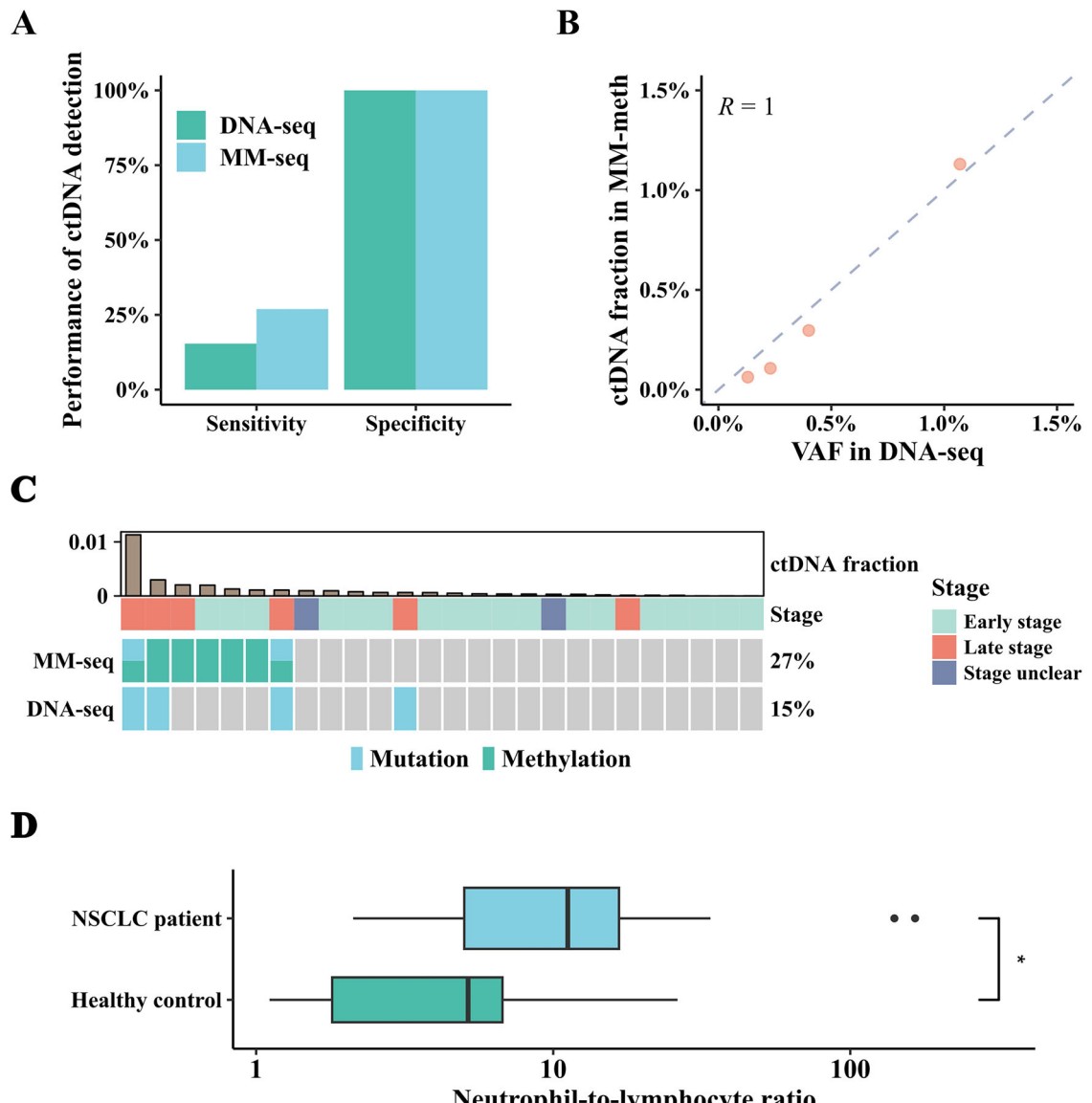

**Figure 4. Detection of ctDNA by MM-seq.**

(A) Sensitivity and specificity of ctDNA detection by standard DNA-seq and MM-seq. Bar plots show detection of ctDNA in 26 NSCLC patients and 13 healthy donors evaluated by DNA-seq and MM-seq. (B) Correlation between VAF of ctDNA evaluated by standard DNA-seq and MM-seq. Scatter plots show VAFs of somatic mutations measured by standard DNA-seq and ctDNA fractions inferred from MM-meth in four NSCLC patients. (C) Detection of ctDNA in twenty-six NSCLC tumor patients using DNA-seq and MM-seq. Each column corresponds to an individual patient, with tumor stage annotated above. The color of the cells indicates the type of information used to identify ctDNA. The upper bar plot represents the ctDNA fraction, inferred from methylation signals detected by MM-seq. (D) Comparison of neutrophil-to-lymphocyte ratio calculated with methylation-inferred immune cell-specific cfDNA between NSCLC patients ($N = 26$) and healthy individuals ($N = 13$) (Student's $t$ test, *$P$ value = 0.049). In boxplots, solid black lines show the median, box edges mark the first and third quartiles, whiskers reach the furthest points within 1.5 times the interquartile range, and outliers beyond this range define the minimum and maximum.

methylation was indicative of biallelic loss of gene function. A significant difference in *BRCA1* promoter methylation was observed between the HRD and HRP groups ($P$ value < 0.01), with *BRCA1* promoter methylation detected in 56.25% (9/16) of HRD patients and in none of the HRP patients (Fig. 5B). This finding suggests a correlation between biallelic loss of function in *BRCA1* introduced by promoter methylation and HRD, further supported by the strong correlation between *BRCA1* promoter methylation levels and HRD scores (Fig. 5C), consistent with previous research

indicating that high-level *BRCA1* methylation is required for HRD in solid tumors (Xu et al, 2024). However, the difference was not significant for *RAD51C* due to the low prevalence of *RAD51C* promoter methylation, which was observed in only one patient in this cohort. Interestingly, while the incidence of biallelic loss in *FANCA* was significantly higher in the HRD group compared to the HRP group (43.75% [7/16] in HRD patients versus 9.09% [1/11] in HRP patients), no correlation was found between *FANCA* promoter methylation and HRD scores (Appendix Fig. S6C). Of the seven

 

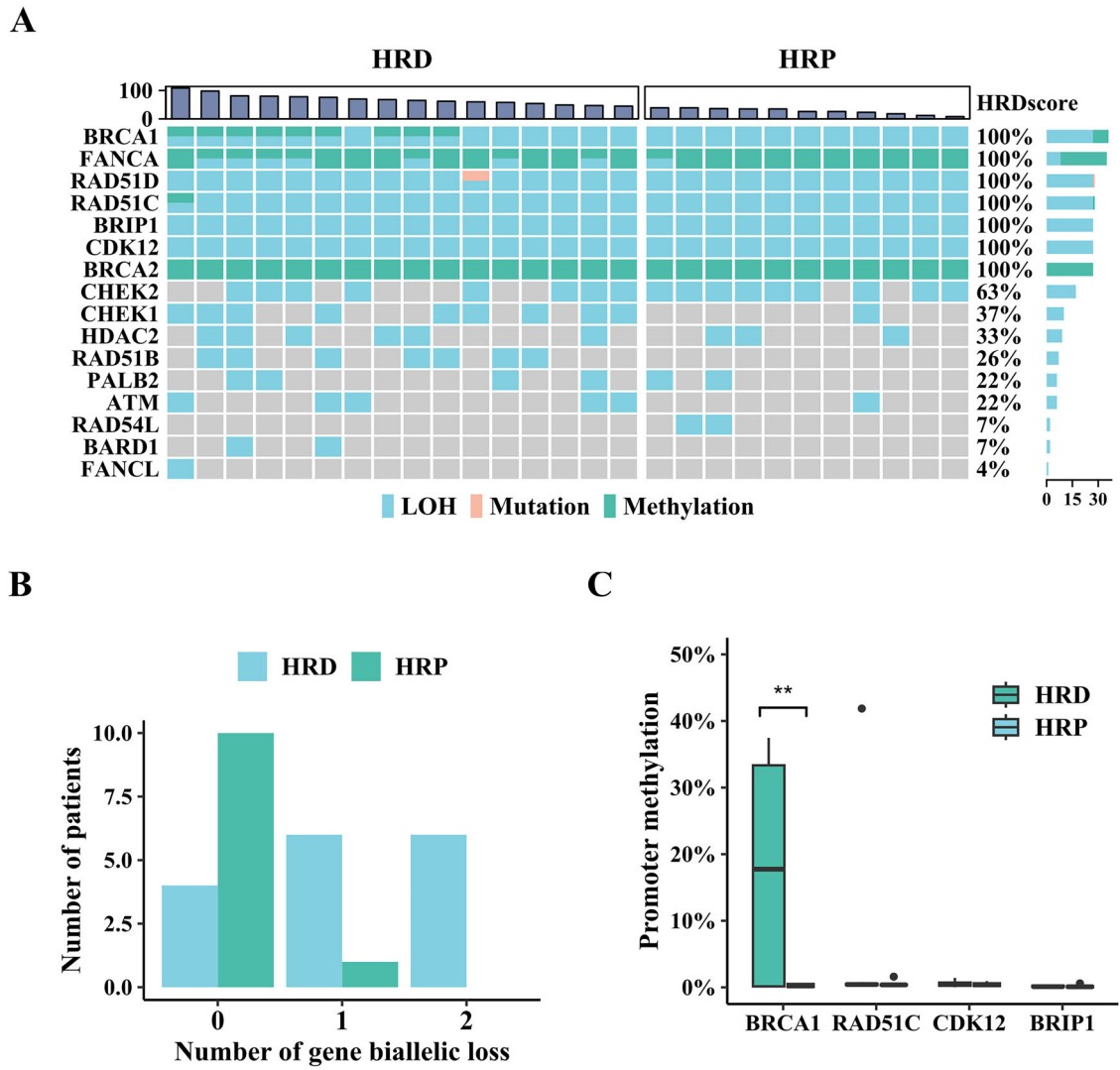

**Figure 5. Detection of biallelic loss of function in HRR genes using MM-seq.**

(A) LOH, mutations, and promoter methylation were identified in FFPE samples from 27 ovarian and breast cancer patients using MM-seq. Each cell represents the variation of a single gene in one sample, with colors indicating the type of variation. The prevalence of altered genes is indicated on the right side of the heatmap. Samples were classified into HRD or HRP groups based on the HRD score. (B) Distribution of HRD and HRP statuses in patients with varying numbers of genes exhibiting biallelic loss. (C) Promoter methylation of *BRCA1*, *RAD51C*, *CDK12*, and *BRIP1* in HRD (N = 16) and HRP (N = 11) groups. LOH for these four genes was observed in all samples. The significance of differences between groups is indicated above the bar plots (Student's *t* test, **P value = 0.0078). In boxplots, solid black lines show the median, box edges mark the first and third quartiles, whiskers reach the furthest points within 1.5 times the interquartile range, and outliers beyond this range define the minimum and maximum.

patients with biallelic *FANCA* loss, five also exhibited *BRCA1* biallelic loss, suggesting the need for further research to clarify the role of *FANCA* promoter methylation in HRD.

## Application of MM co-detection in qPCR

Many genetic testing products are provided as qPCR kits due to their well-established protocols, lower equipment and operational demands, cost-effectiveness, and short turnaround time. Our co-detection method can also be applied to qPCR, referred to as MM-qPCR, in addition to its utilization in high-throughput sequencing. Compared to the standard qPCR protocol, MM-qPCR incorporates an additional initial amplification step to synthesize a mutation-

protecting strand using dmCTP and target-specific primers, followed by bisulfite conversion. We explored this approach in a qPCR-based cancer detection test using 10 ng of gDNA from diluted HCT116 cell line DNA, known to carry the *KRAS* G13D mutation (c.38 G > A) and *SDC2* hypermethylation, at concentrations ranging from 0.05 to 3% (0.05%, 0.1%, 0.5%, 1%, 3%). For comparison, standard qPCR was also used to detect the *KRAS* G13D mutation and *SDC2* hypermethylation, with each detection performed using 10 ng of DNA at the same dilutions as MM-qPCR, following the protocols of the respective kits. Each sample was tested in triplicate for both MM-qPCR and standard qPCR.

MM-qPCR detected the *KRAS* G13D mutation at 0.5%, while standard qPCR detected it at 1% in all three replicates (Fig. 6A,B),

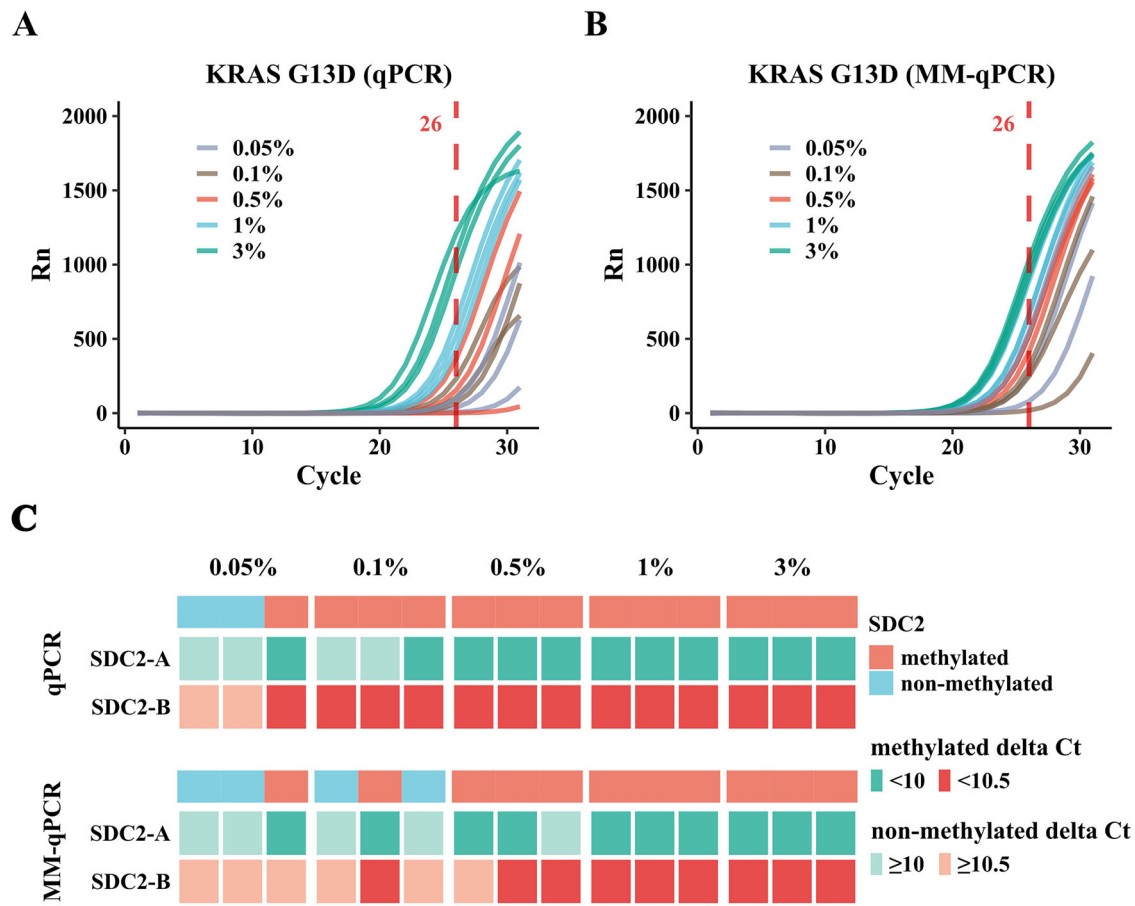

**Figure 6. Application of methylation and mutation co-detection in qPCR (MM-qPCR).**

(A) Detection of *KRAS* G13D mutation using regular qPCR with 3%, 1%, 0.5%, 0.1%, and 0.05% HCT116 gDNA in triplicate. Ct lower than 26 indicates the detection of the mutation. (B) Detection of *KRAS* G13D mutation using MM-qPCR with 3%, 1%, 0.5%, 0.1%, and 0.05% HCT116 gDNA in triplicate. Ct lower than 26 indicates the detection of the mutation. (C), Detection of *SDC2* hypermethylation using standard qPCR and MM-qPCR with 3%, 1%, 0.5%, 0.1%, and 0.05% HCT116 gDNA in triplicate. SDC2-A ΔCt < 10 or SDC2-B ΔCt < 10.5 indicates SDC2 hypermethylation. Source data are available online for this figure.

both reaching the limit of detection of the kit (1%). In addition, both methods were able to detect the mutation as low as 0.05%. For methylation detection, although MM-qPCR was unable to identify *SDC2* hypermethylation at 0.1% in all three replicates, similar to standard qPCR, it did detect *SDC2* hypermethylation at 0.5% in all replicates (Fig. 6C), reaching the detection limit of the kit (1.25%). Both methods were also able to identify methylation as low as 0.05%, as seen with mutation detection. MM-qPCR successfully detected both the *KRAS* G13D mutation and *SDC2* hypermethylation at 0.5%, achieving comparable results to the regular method while requiring only half the sample amount. This validation confirms the suitability of our co-detection strategy for qPCR tests, enhancing its adaptability and precision for broader clinical applications.

## Discussion

In this study, we have described a method that simultaneously captures mutation and methylation information within a single workflow optimized for clinical applications. This streamlined approach eliminates the need for sample splitting, providing an advantage for clinical samples with limited DNA availability while also offering notable cost and time efficiencies (Appendix Table S5). Furthermore, the method is adaptable to multiple molecular detection platforms, making it feasible for integration into various laboratory pipelines. Beyond these methodological benefits, the approach demonstrated performance comparable to state-of-the-art techniques in both methylation analysis and the detection of genomic alterations using targeted sequencing, supporting its potential utility in translational and clinical settings.

For ctDNA detection, methylation patterns tend to provide more informative signals than mutations, potentially offering improved sensitivity (Liu et al, 2018), as demonstrated in our preliminary investigation, where MM-seq modestly improved the ctDNA detection rate pre-treatment cfDNA samples compared to mutation-only approaches by jointly profiling mutations and methylation in a single workflow. Although a gap remains between this level of sensitivity and the reliable detection of tumors in the post-treatment low-ctDNA context, it suggests a possible avenue for improving ctDNA detection sensitivity using limited cfDNA input, without significantly increasing clinical workload. This

improvement could potentially be further enhanced by expanding the methylation marker panel. Moreover, MM-seq may assist in distinguishing between true tumor-derived ctDNA and CHIP without requiring paired WBC testing: ctDNA harboring variants with high frequencies likely exhibit corresponding methylation patterns, and the absence of such patterns suggests the variant is likely CHIP-derived rather than tumor-origin.

Apart from the tumor-derived DNA, cfDNA from other sources, which is often neglected in routine liquid biopsy, also provides significant insights into the tumor dynamics. For instance, immune-related cfDNA provides critical information on the ongoing interactions between the tumor and the immune system. By measuring immune-derived cfDNA, we were able to validate NLR as an indicator of immune homeostasis, distinguishing between healthy individuals and cancer patients. While we observed differences in lymphocyte-derived cfDNA between tumor and healthy control samples, there remains a critical need to increase the resolution of immune cell subsets, such as CD8+ T cells, CD4+ T cells, and CD19+ B cells. These subsets are known to provide more detailed and informative data (Holm et al, 2022; Stankovic et al, 2018), which would enable more comprehensive monitoring of immune dynamics in cancer patients. Moreover, extending the scope of analysis beyond immune cell-specific cfDNA to include broader immune markers, such as circulating cytokine levels (Verschoor et al, 2017), could further enhance the profiling of immune characteristics. This comprehensive profiling may provide deeper insights, facilitating more accurate assessment of immunotherapy responses or resistance, thereby assisting clinical decision-making.

In certain companion diagnostics, such as those for detecting HRD, genomic variation analysis reflects the cumulative effect of past alterations rather than the tumor's current state, which is more accurately reflected by methylation-regulated gene expression. Our investigation on biallelic loss of function in HRR genes revealed that not all patients with HRD exhibit such loss, which may be attributed to reversion mutations or promoter demethylation of HRR genes during tumor progression. These alterations can lead to gene reactivation and subsequent resistance to PARP inhibitors while HRD signatures persist (D'Andrea, 2018; Doig et al, 2023). A similar principle may also contribute to other resistance mechanisms, such as those involving tyrosine kinase inhibitors (Bui et al, 2024; Yang et al, 2022), where integrated analysis of drug target mutations and epigenetic alterations may provide a more comprehensive evaluation of patient conditions and more precise guidance for treatment.

A potential drawback of mixing fragments with variant and methylation information in a single workflow is that they are simultaneously damaged during the bisulfite or enzymatic conversion step, leading to loss of DNA templates and reduced sensitivity in variant detection. To mitigate this, we have adopted an optional solution involving multiple cycles of pre-amplification during the synthesis of the protective copy strand, which increases the likelihood that at least one copy of the original DNA molecules survives the conversion. This pre-amplification step introduces a high load of modified cytosines into the library, which may exceed the capacity of TET enzymes during enzymatic conversion. This likely explains the elevated conversion rates observed when dmCTP was used for protective strand synthesis. Consequently, the modified cytosines used in pre-amplification must possess high resistance to deamination. We selected dpyCTP as the modified dCTP during protective strand synthesis to minimize C-to-T (and G-to-A) errors introduced by the conversion of modified cytosines. Although the minor C-to-T substitution rate in MM-genet could potentially lead to false-positive detection of genomic alterations in ultra-deep sequencing, guanine remains unaltered on the original strand in MM-meth and can serve as a reference to correct false C-to-T substitutions at the complementary loci on the paired copy strand. However, because only a portion of the reads can be paired in the current MM-seq, this limitation both prevents the correction of some false C-to-T substitutions and restricts the detection of low-frequency cis methylation and genetic alterations. We aim to continue screening for more effective analogs and remain committed to improving DNA fragment recovery to further refine this approach.

Another limitation of this study is that the clinical potential of the method was assessed through preliminary studies with limited sample sizes. In particular, the observed sensitivity improvement in ctDNA detection requires further validation in larger cohorts, especially in clinical contexts with low ctDNA levels. TAPS has recently been reported in a diagnostic accuracy study involving a large clinical cohort (Vavoulis et al, 2025), indicating the clinical utility of co-detecting methods. In addition, MM-seq minimizes additional work compared to existing protocols by requiring only one extra experimental step (synthesizing a mutation-protective strand) and one bioinformatics step (allocating primary sequencing data to MM-meth and MM-genet) to achieve co-detection, making the method readily implementable in clinical laboratories. Given this compatibility with existing protocols, we anticipate that validation in large populations could be rapidly achieved by integrating the method into routine clinical testing if necessary.

The MM co-detection strategy offers a flexible and practical tool for the simultaneous characterization of genomic and epigenomic features, with demonstrated potential in various clinical scenarios. This work represents a step toward more practical incorporation of both genomic and epigenomic insights into clinical practice, and we are committed to further optimizing this method for both academic research and clinical diagnostics.

# Methods

**Reagents and tools table**

| Reagent/resource | Reference or source | Identifier or catalog number |
|---|---|---|
| **Experimental models** | | |
| HCT116 | AmoyDx Laboratory | |
| NA12878 | Coriell Institute | |
| 293T | AmoyDx Laboratory | |
| **Recombinant DNA** | | |
| **Antibodies** | | |
| **Oligonucleotides and other sequence-based reagents** | | |
| MM-adapter-1F | This study | Methods |
| MM-adapter-1R | This study | Methods |
| MM-adapter-2 | This study | Methods |
| KRAS-R | This study | Methods |

| Reagent/resource | Reference or source | Identifier or catalog number |
|---|---|---|
| **Chemicals, enzymes, and other reagents** | | |
| AmoyDx® Blood/Bone Marrow DNA Kit | AmoyDx | 8.02.0014 |
| IVD-selected beads | AmoyDx | XW2312001 |
| xGen Prism DNA Library Prep Kit | Integrated DNA Technologies | 10006203 |
| 5-Propynyl-dCTP | TriLink | N-2016 |
| 5-Carboxy-dCTP | TriLink | N-2063 |
| 5-hme-dCTP | TriLink | N-2060 |
| Hieff Canace® High-Fidelity DNA Polymerase | Yeasen | 10135ES80 |
| NEBNext® Enzymatic Methyl-seq Conversion Module | New England Biolabs | E7125L |
| KAPA HiFi HotStart Uracil+ Ready Mix | KAPA Biosystems | KK2802 |
| NEBNext® Enzymatic Methyl-seq | New England Biolabs | E7120L |
| KAPA HiFi HotStart Ready Mix | KAPA Biosystems | KK2502 |
| dmCTP | New England Biolabs | N0356S |
| EZ DNA Methylation-Lightning Kit | Zymo | D5030 |
| *SDC2* Methylation Detection Kit | AmoyDx | 8.01.0231 |
| *KRAS* Mutation Detection Kit | AmoyDx | 8.01.0126 |
| pUC19 plasmid | Sangon | B610005-0050 |
| **Software** | | |
| Seqkit 0.16.1 | | https://bioinf.shenwei.me/seqkit |
| fastp v0.20.0 | | https://github.com/OpenGene/fastp |
| bsbolt v1.6.0 | | https://github.com/NuttyLogic/BSBolt |
| bwa 0.7.17 | | https://github.com/lh3/bwa |
| GATK v4.2.0.0 | | https://gatk.broadinstitute.org/hc/en-us |
| R version 4.2.3 | | https://www.r-project.org/ |
| **Other** | | |
| Illumina NovaSeq | Illumina | |
| Coviras M220 Sonicator | Coviras | |
| SLAN 96P | SLAN | |

## Preparation of cell line DNA

Genomic DNA of NA12878 was procured from Coriell Institute. HCT116 and 293 T cell lines were sourced from in-house cultures and authenticated by short tandem repeat (STR) profiling to rule out cross-contamination. GenomicDNA was isolated using AmoyDx® Blood/Bone Marrow DNA Kit (AmoyDx, 8.02.0014), following the instructions provided in the product manual. All gDNA samples were fragmented using the Coviras M220 Sonicator and purified with IVD-selected beads (AmoyDx, XW2312001). Mixed gDNA samples were prepared by combining HCT116 gDNA and 293 T gDNA to achieve 1% HCT116 (comprising 1 ng of HCT116 gDNA and 99 ng of 293 T gDNA) and 0.1% HCT116 (comprising 0.1 ng of HCT116 gDNA and 99.9 ng of 293 T gDNA).

## Preparation of real-world sample DNA

Genomic DNA and cfDNA from tumor samples were obtained from residual specimens stored in the biobank of Shanghai Xiawei Medical Laboratory Co., Ltd, and cfDNA samples from healthy donors were collected in the laboratory. Informed consent was obtained from all human subjects, specifying that the samples could be utilized for the company's scientific research. The study was carried out at Shanghai Xiawei Medical Laboratory Co., Ltd., in full compliance with the Regulations on Ethical Review of Life Sciences and Medical Research Involving Humans issued by the National Health Commission (NHC) of China. All experiments involving human samples were conducted in accordance with the principles outlined in the World Medical Association (WMA) Declaration of Helsinki and the Department of Health and Human Services Belmont Report.

## Preparation of spike-in control DNA

Fully CpG-methylated pUC19 (0.1 ng/μl, NEB, E7125L) and fully unmethylated lambda DNA (2 ng/μl, NEB) were utilized as spike-in control DNA. A series of dilutions with methylation levels of 5%, 2%, 1%, 0.5%, and 0.2% were prepared by mixing fully CpG-methylated pUC19 with fully unmethylated pUC19 plasmid (Sangon). Subsequently, all spike-in controls underwent fragmentation and purification, following the same procedure as gDNA processing.

## Adapter preparation

Adapter (MM-adapter-1F, 5′-rApp-NNNNNNNNAGAT ATCGG AAGAGCACACGTCTGAACTCCAGTCAC-C3 Spacer-3′, and MM-adapter-1R 5′-CTNNNNNNNNddN-3′), containing 32 types of sequence-fixed UMI, was synthesized (Generay) and annealed in a buffer containing 50 mM Tris·HCl (pH = 8.0) and 10 mM MgCl$_2$. The annealing program was set to 95 °C for 3 min, then ramped down to 10 °C at a rate of 0.1 °C/second. MM-adapter-2 (5′-TTTCCCTACACGACGCTCTTCCGATC-3′) was synthesized (Sangon) for the second ligation step of xGen Prism DNA Library Prep Kit (IDT, 10006203). Both the MM-adapter-1 and MM-adapter-2 had all Cs substituted with mCs, except for the Cs in the 8-base UMI sequences (Appendix Table S1).

## MM-seq, EM-seq, and DNA-seq

For MM-seq, sheared gDNA or cfDNA was combined with spike-in control (fully methylated pUC19 DNA and fully unmethylated lambda DNA) as described in the manual (NEB, E7125L). The library preparation was then performed using the xGen Prism DNA Library Prep Kit (IDT, 10006203) with a modified process. Briefly, DNA underwent end repair, followed by a two-step ligation with MM-adapter-1 and MM-adapter-2 successively. Subsequently, a

mutation-protective pre-PCR was performed using modified dCTP and Hieff Canace® High-Fidelity DNA Polymerase (Yeasen, 10135ES80) to synthesize protected DNA strands with one cycle for gDNA and three cycles for cfDNA. Four modified dCTP analogs, including 5-methyl-dCTP (NEB, N0356S), 5-hydroxymethyl-dCTP (TriLink, N-2016), 5-carboxy-dCTP (Tri-Link, N-2063), and 5-propynyl-dCTP (TriLink, N-2060), were evaluated to identify the most effective cytosine analog for strand protection. All dCTP analogs used had a purity greater than 99% and were confirmed to be free of dCTP contamination, and 5-propynyl-dCTP was selected for use in subsequent experiments. The products were purified with IVD-selected beads (2.5×). The purified products were converted using the NEBNext® Enzymatic Methyl-seq Conversion Module (NEB, E7125L). Finally, a seven cycles pre-PCR using KAPA HiFi HotStart Uracil+ Ready Mix (KAPA Biosystems) and IVD-select Beads purification (1.3×) were conducted to obtain the MM-seq library.

For EM-seq, the NEBNext® Enzymatic Methyl-seq (NEB, E7120L) was used as described in the manual. WGS library preparation was carried out using the xGen Prism DNA Library Prep Kit (IDT, 10006203) following the manufacturer's instructions. All pre-PCR procedures were carried out with seven cycles.

## MM-qPCR, Mutation qPCR, and methylation qPCR

In total, 10 ng of mixed gDNA with weight concentrations of HCT116 at 1% and 0.1% was subjected to a 15-cycle protective linear PCR. This process utilized a target-specific primer of the *KRAS* gene (KRAS-R, 5′-TGTATTAAAAGGTACTGGTGGAGT-3′) and substituted dmCTP (NEB, N0356S) in place of dCTP. The PCR program was set at 95 °C for 2 min, followed by 15 cycles of 95 °C for 2 min, 55 °C for 30 s, and 60 °C for 30 s. The resulting protective product was then treated with EZ DNA Methylation-Lightning Kit (Zymo, D5030) for bisulfite conversion. Following this, 10 µl out of 15 µl converted product was used for testing with the SDC2 Methylation Detection Kit (AmoyDx, 8.01.0231, https://www.amoydx.com/productDetail_24.html), while the remaining 5 µl was used for *KRAS* G13D testing using the KRAS Mutation Detection Kit (AmoyDx, 8.01.0126, https://amoydiagnostics.com/products/amoydx-kras-mutation-detection-kit).

The same 10 ng of mixed gDNA as above was directly subjected to the *KRAS* G13D test with the same mutation detection kit. Simultaneously, another 10 ng of mixed gDNA underwent bisulfite conversion, and the converted product was eluted with 10 µl elution buffer, and then used for *SDC2* methylation testing using the same methylation detection kit as above.

For both MM-qPCR and standard qPCR, *KRAS* G13D mutation was considered positive if the cycle threshold (Ct) value was lower than 26. *SDC2* hypermethylation was confirmed if SDC2-A ΔCt < 10 or SDC2-B ΔCt < 10.5.

## Data downsampling

Libraries prepared with different techniques were sequenced on the Illumina NovaSeq. For each sample replicate of whole-genome sequencing of NA12878, 550 million primary paired-end reads were sampled for MM-seq, resulting in over 165 million read pairs for MM-meth and 300 million read pairs for MM-genet. To ensure comparability across methods, downsampling was performed on

WGS, EM-seq, MM-genet, and MM-meth data using seqkit (https://bioinf.shenwei.me/seqkit/). Specifically, for methylation analysis, EM-seq and MM-meth data were downsampled to 165 million paired-end reads, while for mutation analysis, WGS and MM-genet data were downsampled to 300 million paired-end reads.

## Data processing

For MM-seq, UMIs were generated by extracting the first 8 bases from both reads of the raw read pairs. The fastq files were trimmed and filtered using fastp v0.20.0 (https://github.com/OpenGene/fastp) with the parameters -q 20 -u 30 -n 10. Subsequently, reads with UMIs subjected to C-to-T (G-to-A) conversion were assigned to the methylation analysis, while those that matched the primary designed sequences were designated for mutation analysis. For the methylation analysis, reads were aligned to the converted genome comprising of hg19 human reference genome and spiked-in control sequences using bsbolt v1.6.0 (https://github.com/NuttyLogic/BSBolt). Duplicate reads were removed using the MarkDuplicates from the GATK suite v4.2.0.0 (https://gatk.broadinstitute.org/hc/en-us). The resulting deduplicated BAM files were then utilized for methylation calling using bsbolt. To analyze genetic alterations, reads from MM-genet were aligned to the standard hg19 human genome using bwa 0.7.17 (https://github.com/lh3/bwa). To identify SNP, duplicate removal was conducted using the MarkDuplicates, and recalibration was performed on the deduplicated BAM files using GATK. SNPs were called using the HaplotypeCaller tool from GATK, with subsequent filtering for variants meeting specific criteria: a minimum coverage of 10×, at least two supporting reads, and a minimum VAF of 0.15. For the detection of copy number variations, the genome was divided into non-overlapping 10 kb bins, and the coverage within each bin was gathered. Bins with less than 1× coverage were filtered out. The copy ratio for each bin was calculated by dividing its coverage by the median coverage of all bins across the genome, followed by a correction based on GC content. Any segment with copy number exceeding 3.5 or falling below 0.5 was identified as a potential candidate for copy number variation. To identify somatic mutations in target-enrichment deep sequencing, base calibration and duplicate removal were performed by clustering consensus reads using an in-house script. Somatic variants were called with VarScan, and synonymous variants, as well as variants within introns and UTR regions, were excluded. Variants with fewer than five supporting reads or a VAF less than 0.1% were also excluded to minimize false positives. Variants that were detected in both cfDNA and paired WBC samples were discarded to rule out CHIP. C-to-T or G-to-A variants were also left out to avoid falsely called variants introduced by deamination of modified cytosines.

For DNA-seq, UMIs were extracted from the raw data and fastq files were processed using fastp as described for MM-seq, applying the same criteria. Trimmed data underwent procedures similar to the mutation analysis in MM-seq, with the exception that C-to-T and G-to-A variants were retained for somatic mutation identification. For EM-seq, UMI extraction was not performed, as no UMIs are present in the adapter in the standard EM-seq protocol. EM-seq data underwent a process similar to the methylation analysis in MM-seq, starting from filtering with fastp. Duplicate rates were assessed using GATK MarkDuplicates for methylation analysis,

while for somatic mutation analysis, duplicate rates were calculated by considering reads with the same genomic coordinates within five base pairs and UMI sequences with no more than 1 mismatch as duplicates.

## Allele-specific methylation

Allele-specific methylation candidates in NA12878 were gathered from previous reports. To target selected SNPs, methylation and sequence probes were designed to capture the MM-seq library. DNA fragments encompassing the SNPs, along with their outer mapping coordinates and flags, were extracted from the BAM files of MM-genet and MM-meth using pysam. Fragments from MM-genet and MM-meth were considered to originate from the same molecule if they met the following criteria: (1) identical mapping coordinates, (2) paired original and converted UMIs, and (3) consistent strand origin, either from the Watson strand (MM-genet read flag equals 99 or 147, and MM-meth read flag equals 67 or 131) or the Crick strand (MM-genet read flag equals 83 or 163, and MM-meth read flag equals 115 or 179). Successfully paired fragments were further categorized into two groups by pairing read IDs with pileup results: one group containing the wild-type allele and the other containing the mutant allele. For each group, the number of methylated and unmethylated cytosines in the CpG context around the SNP loci was quantified. Methylation differences between the alleles were then compared using Fisher's exact test.

## ctDNA detection

Detection of ctDNA was evaluated using standard DNA-seq as a benchmark with the LC27 panel, a targeted sequencing panel comprising 27 genes associated with NSCLC, designed specifically for MRD detection in NSCLC patients. This panel builds on the earlier LC10 panel by incorporating additional drug-resistant and recurrently mutated genes in NSCLC. The design process involved sequentially adding exons from a broader pan-cancer panel (Master Panel, AmoyDx) to the LC10 panel. After each addition, the proportion of patients with at least one detectable mutation in both the CHOICE and TCGA lung cancer cohorts was evaluated. The exon that led to the most significant increase in this proportion was retained. This iterative process continued until the proportion of patients with at least one detectable mutation surpassed 85% in both cohorts. The final LC27 panel was combined with methylation probes targeting eight NSCLC-associated methylation clusters and seventeen immune-specific methylation markers to capture MM-seq libraries prepared from the same samples. Somatic mutations were identified for both DNA-seq and MM-seq as previously described. For methylation analysis, fragments overlapping the targeted regions of the eight methylation markers were extracted for downstream analysis. Fragments covering fewer than five CpG sites were excluded. Given that CpGs within all markers are highly hypermethylated in NSCLC, a fully CpG-methylated fragment was considered to be tumor-derived, specifically a ctDNA fragment. The number of ctDNA fragments was then counted and divided by the total number of available fragments to calculate the ctDNA fraction. A sample was classified as ctDNA-positive if any somatic mutation was detected, or if the methylation-derived ctDNA fraction exceeded 0.1%, provided that at least three ctDNA

fragments were present and the total number of cfDNA fragments was no fewer than 3000.

## Immune-derived cfDNA

Genomic DNA and cfDNA samples from 10 healthy donors were subjected to MM-seq and then captured with probes targeting 17 established methylation markers specific to seven immune cell types. DNA fragments utilized for calculating the fraction of each immune cell type were selected based on coverage of all CpG sites within the genomic regions corresponding to the respective methylation markers. The fraction of each immune cell type was determined by averaging the proportion of fragments containing the requisite number of unmethylated CpG sites for all markers associated with that cell type. To calibrate the immune-specific DNA fractions, a regression model was generated by fitting the immune-specific gDNA fractions from five samples against the immune cell proportions obtained from reported complete blood count (CBC) data in a reference cohort of healthy individuals. This model was subsequently applied to calculate the immune-specific gDNA fractions for the remaining five samples and the cfDNA fractions for all ten cfDNA samples. Finally, the DNA fractions were normalized so that the sum of the fractions of all immune cell types equaled 1 for each sample. Immune cell proportions measured by FCM were normalized according to CBC measurements. Specifically, the sum of FCM-derived lymphocyte proportions was adjusted to match the CBC-derived lymphocyte proportions, and the sum of neutrophil, eosinophil, and monocyte proportions measured by FCM was similarly adjusted to align with the CBC-derived proportions.

## Biallelic loss

The HumanMethylation450 methylation profiles and RNA-seq gene expression data for 690 TCGA breast cancer samples were obtained from UCSC Xena (https://xenabrowser.net/datapages/). The correlation between CpG methylation and gene expression was calculated for each CpG site annotated within 16 HRR genes (*ATM, BARD1, BRCA1, BRCA2, BRIP1, CDK12, CHEK1, CHEK2, FANCA, FANCL, HDAC2, PALB2, RAD51B, RAD51C, RAD51D, RAD54L*). A CpG site was defined as expression-correlated if it met the following criteria: Pearson's correlation coefficient was less than $-0.15$, the CpG site was located within the gene promoter region (defined as 1000 bp upstream to 750 bp downstream of the transcription start site), and there were more than five CpG sites within 75 bp flanking region of the CpG site. The CpG candidates were merged into a cluster when their respective 100 bp flanking regions exhibited overlap. If a CpG site did not have an overlapping flanking region with other CpG candidates, a 75 bp flanking region was defined as an expression-correlated CpG cluster. The methylation status of a gene was determined based on the mean beta value of CpG sites within the gene's expression-related CpG cluster. A gene was classified as hypermethylated if the mean beta value of the corresponding CpG sites exceeded 0.07. To identify LOH, normalized depth and B-allele frequency (BAF) of probes were used to estimate the gene's normalized depth and BAF. The copy number (M) of the major allele and the copy number (N) of the minor allele were then determined using a maximum posterior probability model that incorporated tumor purity and ploidy. LOH

was identified if N equaled zero and M was no less than one for a gene. Biallelic loss of function of a gene was identified if any of the following criteria were met: homozygous deletion, LOH combined with promoter methylation, LOH combined with deleterious mutation, and two deleterious mutations.

### Statistical analysis

Sample sizes were not predetermined by statistical methods; instead, they were selected based on sample availability, prior literature, and professional judgment. No randomization was employed in the study, and no samples were excluded from the analysis. Blinding was not implemented during either the experimental procedures or data analysis.

All statistical analyses were conducted using R version 4.2.3. Pearson correlation coefficients were used to evaluate the CpG methylation correlation and VAF correlation, as well as the correlation between HRR gene promoter methylation and HRD score. Fisher's exact test was applied to evaluate allele-specific methylation and to examine the association between biallelic loss of function in HRR genes and HRD status. Comparisons of cfDNA-inferred immune characteristics between cancer patients and healthy controls were performed using Student's $t$ test.

## Data availability

Data described in this study are available from Genome Sequence Archive for Human through accession code PRJCA030061 (https://ngdc.cncb.ac.cn/gsa-human/browse/HRA008602). All data are available in the main text or the supplementary materials.

The source data of this paper are collected in the following database record: biostudies:S-SCDT-10_1038-S44321-025-00259-7.

## Peer review information

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

## Acknowledgements

This work was supported by Amoy Diagnostics, where the authors are employed. The company provided funding for the research and publication process as part of its research and development program.

## Author contributions

**JIyan Yu**: Conceptualization; Investigation; Methodology; Writing—original draft. **Chunhe Yang**: Formal analysis; Investigation; Visualization; Writing—original draft. **Xintao Zhu**: Investigation. **Zhankun Wang**: Investigation. **Boping Xu**: Formal analysis. **Ye Cai**: Investigation. **Jingbo Zhao**: Investigation. **Ruijian Guo**: Formal analysis. **Wuzhou Yuan**: Formal analysis. **Jianqing Wang**: Formal analysis. **Bohao Dong**: Investigation. **Frank Ron Zheng**: Supervision; Writing—review and editing. **Shuang Yang**: Conceptualization; Supervision; Writing—review and editing.

Source data underlying figure panels in this paper may have individual authorship assigned. Where available, figure panel/source data authorship is listed in the following database record: biostudies:S-SCDT-10_1038-S44321-025-00259-7.

## Disclosure and competing interests statement

The authors are employees of Amoy Diagnostics and have filed a CNIPA patent on MM-seq, application number CN202311842967.X, Amoy Diagnostics Co., Ltd, filed 28 December 2023. The remaining authors declare no competing interests.

