## [Peer Review File · EMBO Molecular Medicine]

A DNA alteration and methylation co-detection method for clinical purpose

Jlyan Yu, Chunhe Yang, Xintao Zhu, Zhankun Wang, Ye Cai, Jingbo Zhao, Ruijian Guo, Wuzhou Yuan, Jianqing Wang, Bohao Dong, Boping Xu, Frank Zheng, and Shuang Yang

Corresponding authors: Shuang Yang (shuang.yang@amoydx.com) , Frank Zheng (frank.zheng@amoydx.com)

Review Timeline:

Submission Date:	28th Nov 24
Editorial Decision:	10th Jan 25
Revision Received:	7th Apr 25
Editorial Decision:	29th Apr 25
Revision Received:	12th May 25
Authors' Correspondence:	22nd May 25
Accepted:	22nd May 25

Editor: Lise Roth

Transaction Report:

10th Jan 2025

Dear Dr. Zheng,

Thank you for the submission of your manuscript to EMBO Molecular Medicine, and please accept my apologies for the delay in getting back to you during this busy time of the year. We have now heard back from the referees who reviewed your manuscript. As you will see from the reports below, the referees acknowledge the novelty and interest of the study and are overall supporting publication of your work pending appropriate revisions.

Addressing the reviewers' concerns in full will be necessary for further considering the manuscript in our journal, and acceptance of the manuscript will entail a second round of review. EMBO Molecular Medicine encourages a single round of revision only and therefore, acceptance or rejection of the manuscript will depend on the completeness of your responses included in the next, final version of the manuscript. For this reason, and to save you from any frustrations in the end, I would strongly advise against returning an incomplete revision.

We are expecting your revised manuscript within three months, if you anticipate any delay, please contact us.

We require:

4) A .docx formatted letter INCLUDING the reviewers' reports and your detailed point-by-point responses to their comments. As part of the EMBO Press transparent editorial process, the point-by-point response is part of the Review Process File (RPF), which will be published alongside your paper.

5) A complete author checklist, which you can download from our author guidelines (<https://www.embopress.org/page/journal/17574684/authorguide#submissionofrevisions>). Please insert information in the checklist that is also reflected in the manuscript. The completed author checklist will also be part of the RPF.

6) All Materials and Methods need to be described in the main text using our 'Structured Methods' format. According to this format, the Methods section includes a Reagents and Tools Table (listing key reagents, experimental models, software and relevant equipment and including their sources and relevant identifiers) followed by a Methods and Protocols section describing the methods, ideally using a step-by-step protocol format. The aim is to facilitate adoption of the methodologies across labs. Please download and fill our Reagents and Tools Table template (.docx), which you can find in our author guidelines: <https://www.embopress.org/page/journal/14693178/authorguide#structuredmethods>.

<https://www.embopress.org/doi/10.15252/msb.20178071>

7) Please note that all corresponding authors are required to supply an ORCID ID for their name upon submission of a revised manuscript.

8) It is mandatory to include a 'Data Availability' section after the Materials and Methods. Before submitting your revision, primary datasets produced in this study need to be deposited in an appropriate public database, and the accession numbers and database listed under 'Data Availability'. Please remember to provide a reviewer password if the datasets are not yet public (see <https://www.embopress.org/page/journal/17574684/authorguide#dataavailability>).

9) For data quantification: please specify the name of the statistical test used to generate error bars and P values, the number (n) of independent experiments (specify technical or biological replicates) underlying each data point and the test used to calculate p-values in each figure legend. The figure legends should contain a basic description of n, P and the test applied. Graphs must include a description of the bars and the error bars (s.d., s.e.m.). Please provide exact p values.

10) Our journal encourages inclusion of *data citations in the reference list* to directly cite datasets that were re-used and obtained from public databases. Data citations in the article text are distinct from normal bibliographical citations and should directly link to the database records from which the data can be accessed. In the main text, data citations are formatted as follows: "Data ref: Smith et al, 2001" or "Data ref: NCBI Sequence Read Archive PRJNA342805, 2017". In the Reference list, data citations must be labeled with "[DATASET]". A data reference must provide the database name, accession number/identifiers and a resolvable link to the landing page from which the data can be accessed at the end of the reference. Further instructions are available at .

11) We replaced Supplementary Information with Expanded View (EV) Figures and Tables that are collapsible/expandable online. A maximum of 5 EV Figures can be typeset. EV Figures should be cited as 'Figure EV1, Figure EV2' etc... in the text and their respective legends should be included in the main text after the legends of regular figures.

12) The paper explained: EMBO Molecular Medicine articles are accompanied by a summary of the articles to emphasize the major findings in the paper and their medical implications for the non-specialist reader. Please provide a draft summary of your article highlighting

13) Author contributions: CRediT has replaced the traditional author contributions section because it offers a systematic machine readable author contributions format that allows for more effective research assessment. Please remove the Authors Contributions from the manuscript and use the free text boxes beneath each contributing author's name in our system to add specific details on the author's contribution. More information is available in our guide to authors.

Please also suggest a visual abstract to illustrate your article as a PNG file 550 px wide x 300-600 px high. A cropped portion of this image will serve as thumbnail for the table of content on our webpage.

16) As part of the EMBO Publications transparent editorial process initiative (see our Editorial at <http://embomolmed.embopress.org/content/2/9/329>), EMBO Molecular Medicine will publish online a Review Process File (RPF) to accompany accepted manuscripts.

In the event of acceptance, this file will be published in conjunction with your paper and will include the anonymous referee reports, your point-by-point response and all pertinent correspondence relating to the manuscript. Let us know whether you agree with the publication of the RPF and as here, if you want to remove or not any figures from it prior to publication. Please note that the Authors checklist will be published at the end of the RPF.

I look forward to receiving your revised manuscript.

Yours sincerely,

Lise Roth

***** Reviewer's comments *****

Referee #1 (Remarks for Author):

In this study, Yu and colleagues describe the development of MM-seq, a method tailored to infer both genetic and epigenetic information from the same DNA sample. The method is subsequently applied to a set of cfDNA and DNA samples for sequencing and qPCR, yielding the anticipated outcomes. Despite a few open questions, the rationale for this work seems sound and of potential interest to the community.

1. I have the following questions:

2. In the introduction, the authors describe issues with currently available methods for joint profiling of DNA methylation and genetic information. I am unsure what issues the authors ascribe to Methyl-SNP-seq.
3. The difference in cytosine methylation levels detected in CHH and CHG contexts for EM-seq and MM-seq are quite different. How do the authors explain this discrepancy?
4. As the authors mention, a key metric for the implementation of MM-seq is the deamination rate of modified cytosines. The results suggest that the deamination rate for 5mC is higher than for 5pyC. However, an alternative explanation is reagent impurity (e.g. d5mCTP being contaminated with dCTP). Can the authors exclude this confounder? This is relevant as the 5mC conversion rate mentioned is notably higher than what was reported in the original EM-seq paper (0.9%).
5. MM-seq relies on the amplification of deaminated DNA as well as of DNA that is protected from deamination. As the former has a markedly lower GC%, the amplification efficiency is anticipated to be higher. How is the ratio of MM-genet to MM-meth altered with varying PCR cycle numbers? This ratio will be artificially altered by the 3 cycles of complementary strand synthesis the authors describe. Will various input amounts (each with their PCR cycle number) require a different number of cycles of complementary strand synthesis?
6. In the target capture experiments described, the number of reads captured from MM-meth is an order of magnitude lower than from MM-genet. Can the authors explain this striking difference?
7. The discussion is comprehensive but needs to be toned down. The authors are affiliated to a commercial company, and in its current state, the discussion feels somewhat overblown and in part a marketing exercise.
8. Can the authors clarify how duplication rates were calculated? The use of UMIs in this context is unclear from the methods section.

Referee #2 (Remarks for Author):

This manuscript has demonstrated a new method called MM-seq to profile the genetic and methylation information from DNA by introducing modified dCTP protective from base conversion. Furthermore, the authors have shown potential clinical application of this method for cancer detection and monitoring based on targeted sequencing or qPCR. The main novelty of the technique is the simultaneous detection of mutations and methylation from the same DNA molecule while being compatible with very low quantity of samples. The detailed concerns are listed as follows:

Major points

1. What's the proportion of MM-meth and MM-genet reads? After 3 cycles of complementary strand synthesis for both Watson and Crick strands with modified dCTP amplification (line 98-99), the proportion of template remained for MM-meth and MM-genet profiling should be around 1:7 if assuming 100% PCR amplification efficiency. Will the difference lead to sequencing imbalance especially for methylation profiling? For genomic DNA, reads from MM-meth is around 2/3 of that from MM-genet (line 142-144). However, in the Method session, the pre-amplification cycles is 1 for gDNA and 5 for cfDNA (line 467-468). Please elaborate the effect of amplification cycles on the sequencing imbalance, especially for low input samples, such as cfDNA?
2. Is the high C to T, G to A rate in MM-seq compared to WGS mainly from modified base conversion (line 167-168 and Figure EV3B)? Even though the overall rate of C to T mutations is low, but it is substantially higher than WGS. Exclusion of C to T and G to A mutation profiling (Method Session, line 529-530) could largely limit the application of the MM-seq method.
3. The authors should also generate a mutational spectrum to compare the different types of mutations from MM-seq and WGS more clearly.
4. The authors have reported the duplication rate of MM-meth and MM-genet from target sequencing with 15ng of cfDNA input (line 196-199). Clinical available cfDNA could be as low as several nanogram or even at sub-nanogram level. It's important to evaluate the sequencing parameters, such as duplication level and unique sequencing depth with lower cfDNA input since the authors have prepared library with 2ng and 10ng of cfDNA as input (line 185-187).
5. How much cfDNA has been obtained from the 10 NSCLC patients for comparison of MM-seq and standard DNA-seq (line 202-205)?
6. Are the sequencing depth and duplication level comparable between MM-seq and standard DNA-seq for MAF profiling (line 205-207)?
7. The authors have demonstrated the ability of MM-seq to detect allele-specific methylation level with targeted profiling for the selected 8 CpGs around 5 SNPs (line 224-225)? Are there any false positives if not targeting known ASM regions?
8. The superior performance for NSCLC detection of MM-seq compared to DNA-seq seems to be completely contributed by methylation profiling (line 269-273, Figure 4C). Using mutations it seems MM-seq was totally inadequate in detecting NSCLC samples. As such it is not meaningful to compare MM-seq with targeted DNA-seq for cancer detection. A comparison should be made between MM-seq and targeted methylation sequencing for cancer detection.
9. The results from qPCR profiling of low-level SDC2 hypermethylation seems to be not highly repeatable for both MM-qPCR and standard qPCR (Figure 6). The authors should repeat at least 3 times for each qPCR samples for both mutation and methylation profiling. It's necessary to provide results from more repeats to prove the performance consistency. On the other hand, the authors should also include performance comparison of MM-qPCR with other commercial kits that measure KRAS G13D mutation and SDC2 methylation, instead of only with their own kits.
10. Since a major goal of the method is for clinical applications, more discussion should be made in terms of the additional time/cost, in terms of laboratory work and bioinformatics, that may be required to implement MM-seq in a clinical lab.

Other minor points:

11. Line 153 should be describing Fig 2A.
12. Typo in line 293 with extra "s": "formalin-fixed paraffin-embedded (FFPE) samples from ovarian and breast cancer patients s"
13. Line 332-333 should be short turnaround time, not "short turnaround times"

Referee #3 (Comments on Novelty/Model System for Author):

Technical Quality (including statistical analysis): Medium

The technical quality of the study is generally good; however, there are notable concerns that warrant further attention. These include the potential for false-positive mutation detection, limited sample sizes, and the lack of replication in key experiments. Thus, while the methodology demonstrates promise, further rigorous testing is required to ensure its robustness.

Novelty: High

The study presents a highly novel approach, specifically the MM-seq method for ctDNA detection, which offers a significant advancement in diagnostic sensitivity and has the potential to enhance therapeutic decision-making. This innovation could provide a valuable contribution to the field, though its clinical applicability requires further validation.

Medical Impact: Medium

The potential medical impact of the study is considerable, as the proposed methodology could improve the detection and monitoring of ctDNA in clinical settings. However, the current study's limited sample size and the absence of validation with tumor tissue sequencing reduce the immediate applicability of the findings to clinical practice. Further studies with larger cohorts are needed to fully assess the clinical utility of the approach.

Adequacy of Model System: Adequate

The model system employed in this study is adequate in demonstrating the feasibility of MM-seq for ctDNA analysis. However, several limitations need to be addressed to improve its clinical relevance. These include concerns regarding the accuracy of mutation detection. Further validation with known mutations and larger clinical cohorts would significantly enhance the robustness of the results shown.

Referee #3 (Remarks for Author):

The reviewed article, "A DNA alteration and methylation co-detection method for clinical purpose," addresses a critical gap in current genomic and epigenomic diagnostics. Traditional methodologies often require separate workflows for DNA mutation and methylation detection, which can be resource-intensive and impractical, particularly in clinical contexts with limited sample availability.

The authors propose a novel co-detection method (MM-seq) designed to overcome these limitations by integrating the detection of mutations and methylation into a streamlined workflow. By utilizing a mutation-protective strand synthesis with modified deoxycytidine triphosphates (dCTPs), this approach offers enhanced sensitivity, specificity, and adaptability to various diagnostic platforms, including next-generation sequencing (NGS) and qPCR.

The article is well-written and presents an interesting exploration of a clinically relevant topic. While the detailed methodology and comprehensive analysis highlight the innovation's potential to improve diagnostic efficiency and sensitivity in circulating tumor DNA (ctDNA) detection, further experiments are needed to rule out false positive mutation detection. Additionally, more robust results are required to demonstrate the extrapolation of these findings to clinical settings and their impact on therapeutic decision-making.

Specific points where the article should be revised to meet the standards of a high-impact journal such as EMBO Molecular Medicine will be outlined in the following sections.

MAJOR POINTS

INTRODUCTION

Page 2, lines 67-68:

- The authors state, "Furthermore, most existing methods lack sufficient clinical validation to confirm their efficacy in diagnostics." This is a significant claim and should be substantiated with references to relevant studies or data.

PRINCIPLE OF MM-SEQ

Page 4, line 126:

- The authors note that "Libraries using dpyCTP demonstrated the highest proportion of perfectly matched UMIs (95.45%, Supplementary Fig EV1A)." This percentage seems not optimum for ultrasensitive mutation tests for MRD detection. Concerns arise regarding whether the method introduces mutations in UMIs or in the inserts during bisulfite conversion. This issue is not tested with the whole-genome sequencing experiments provided. The authors should perform whole-exome sequencing or gene panel sequencing at optimal depth using DNA with known mutations to evaluate their method's accuracy and the rate of false-positive mutations introduced.

ASSESSMENT OF APPLYING MM-SEQ IN CLINICAL PRACTICE

Page 5, lines 192-196:

- The authors mention, "These libraries were captured using a hybrid panel designed for ctDNA assessment, incorporating methylation probes targeting approximately 21 kb of the genome and sequence probes targeting approximately 38 kb (detailed in the methods section under ctDNA detection). MM-seq generated a mean of 2.1 million paired-end reads for MM-meth and 20.2 million paired-end reads for MM-genet..." It is unclear why such a significant difference in paired-end reads exists between methylation and sequence probes if the targeted regions are of similar size. Clarification or additional data addressing this discrepancy is needed.

Figure 3B:

- The specific mutant allele frequency (MAF) values should be displayed.

Page 6, lines 228-230:

- "By pairing reads from MM-genet and MM-meth based on mapping positions and UMI sequences, 36.28% of fragments in MM-meth found their pairs in MM-genet and were allocated to discrete alleles of the target SNPs." This percentage of paired meth and genet reads seems low for reliable cis methylation and genetic alteration detection in MRD protocols. This limitation should be explicitly acknowledged and discussed.

MM-SEQ ENHANCES TUMOR DETECTION AND MONITORING

Page 7, lines 264-268:

- The authors report a higher ctDNA-positive detection rate with MM-seq (26.92%) compared to traditional methods (15.38%). However, they should clarify the criteria used to distinguish tumor-derived mutations from artifacts or false positives. Tumor tissue sequencing should be included to confirm the somatic nature of detected mutations.
- The clinical sample size (26 NSCLC patients) is insufficient for robust conclusions. At least 50 cfDNA samples should be analyzed to strengthen the validity of the study.

MM-SEQ AIDS IN THERAPEUTIC GUIDING

Page 7, lines 295-296:

- The authors categorize samples into HRD (N=16) and HRP (N=11) groups. The sample size is too small to draw meaningful conclusions. Both groups should include at least 30 samples.
- Clinical characteristics of the patients should be presented in table/s. It is crucial to specify whether the breast cancer patients

included are early-stage or metastatic. The methods section should confirm adherence to ethical standards.

MM-QPCR VALIDATION

Page 9, lines 349-353:

- The authors compare MM-qPCR results with standard methods but only perform the experiments once. Replicates should be performed in triplicate, and additional dilutions between 1% and below 0.1% should be explored. Testing with real plasma samples from tumor patients is recommendable to assess the methodology's practical efficiency.

**** Reviewer's comments ****

Referee #1 (Remarks for Author):

In this study, Yu and colleagues describe the development of MM-seq, a method tailored to infer both genetic and epigenetic information from the same DNA sample. The method is subsequently applied to a set of cfDNA and DNA samples for sequencing and qPCR, yielding the anticipated outcomes. Despite a few open questions, the rationale for this work seems sound and of potential interest to the community.

We greatly appreciate the reviewer's positive evaluation of our work. We are glad that the rationale and potential impact of MM-seq were deemed sound. We have carefully

reviewed the comments and addressed the questions in detail. The feedback has been invaluable in refining our presentation and enhancing the clarity of our description. Below, we address each concern raised:

1. I have the following questions:

2. In the introduction, the authors describe issues with currently available methods for joint profiling of DNA methylation and genetic information. I am unsure what issues the authors ascribe to Methyl-SNP-seq.

Response: We appreciate the reviewer for bringing up this question. The primary limitation of Methyl-SNP-seq is its possibly reduced ability to reliably detect low-frequency somatic mutations, which are critical for clinical applications. This limitation arises due to its DNA fragment recovery efficiency. According to Supplemental Table 6 in the original Methyl-SNP-seq paper (PMID 36332968), it achieved a mean coverage of up to 12 across the exome region with 11 million deconvoluted reads. In contrast, MM-seq achieved a mean coverage of 19 over the exome region using only 7 million paired-end MM-genet reads, demonstrating significantly improved performance. Additionally, Methyl-SNP-seq modifies the structure of DNA fragments during library preparation. While this structural alteration enhances the detection of cis-methylation and mutation co-alterations, it also restricts the method's adaptability across different molecular platforms, like qPCR or ddPCR.

3. The difference in cytosine methylation levels detected in CHH and CHG contexts for

EM-seq and MM-seq are quite different. How do the authors explain this discrepancy?

Response: We appreciate the reviewer for noticing this difference. Although there is indeed a notable discrepancy in cytosine methylation levels in the CHH and CHG contexts of the NA12878 sample between MM-seq (0.11%) and EM-seq (0.43%), these values fall within the range reported in the EM-seq study (0.1% to 0.6%, PMID 34140313). We believe this discrepancy may arise from the read assignment process in MM-seq. The segregation of reads requires a perfect match in UMI sequences, which excludes reads with unconverted cytosines in the UMI from MM-meth. This exclusion process may remove reads that tend to carry unmethylated cytosines that are not successfully converted, leading to a higher accuracy in detecting unmethylated cytosines and reducing false-positive methylation in the CHH and CHG contexts for MM-meth. This is also consistent with the conversion sensitivity evaluation in fully unmethylated lambda DNA (99.89% in MM-seq vs. 99.49% in EM-seq, **lines 172-174**).

4. As the authors mention, a key metric for the implementation of MM-seq is the deamination rate of modified cytosines. The results suggest that the deamination rate for 5mC is higher than for 5pyC. However, an alternative explanation is reagent impurity (e.g. d5mCTP being contaminated with dCTP). Can the authors exclude this confounder? This is relevant as the 5mC conversion rate mentioned is notably higher than what was reported in the original EM-seq paper (0.9%).

Response: We appreciate the reviewer's concern regarding the potential influence of reagent impurities, on the observed deamination rates. To address this point, we inquired

the preparation of dmCTP we used in the evaluation. The d5mCTP was synthesized by attaching triphosphoric acid to 5-Methyl-2-deoxycytidine, and the impurities primarily consist of degradation products of phosphoric acid, which do not exceed 1%. It is hardly possible that dmCTP was contaminated with dCTP and we can confidently exclude this as a confounder in our results.

Regarding the higher 5mC conversion rate observed in our MM-seq study compared to the original EM-seq study, we believe this difference can be primarily attributed to the differing quantities of 5mC present in the libraries prepared for each study. The original EM-seq paper reported that 5mC constitutes approximately 2.675% of all cytosines in unconverted DNA. Given that cytosines account for about 25% of all bases, the 200 ng of DNA used in that study contained roughly 1.34 ng of 5mC. In contrast, although we used only 30 ng of DNA to prepare our MM-seq libraries, the pre-amplification step introduced approximately 7.5 ng of 5mC into the library (not including 5mC from the adapter), as all cytosines in the copy strand were synthesized using d5mCTP. Known that the TET kit used in the EM-seq study can accommodate up to 200 ng of DNA (see page 3 in instruction:

<https://www.neb.cn/zh-cn/-/media/nebus/files/manuals/manuale7125.pdf?rev=9d2df4556f37428e9a6d9e2411d40ac6&hash=608DCE3068C9704B7712E28B1505604B>), the amount of 5mC in our libraries exceeded this capacity, whereas the EM-seq libraries did not. This discrepancy likely accounts for the higher 5mC conversion rate observed in our MM-seq study. This is also why we select the APOBEC deamination-resistant analog to minimize the effects of incomplete protection by TET.

5. MM-seq relies on the amplification of deaminated DNA as well as of DNA that is protected from deamination. As the former has a markedly lower GC%, the amplification efficiency is anticipated to be higher. How is the ratio of MM-genet to MM-meth altered with varying PCR cycle numbers? This ratio will be artificially altered by the 3 cycles of complementary strand synthesis the authors describe. Will various input amounts (each with their PCR cycle number) require a different number of cycles of complementary strand synthesis?

Response: We thank the reviewer for this insightful comment. In response, we conducted an experiment using 30 ng of gDNA from NA12878 with 12 replicates. The products from these replicates were pooled together after conversion and then split into 12 equal parts, each subjected to different cycles of index PCR (3, 5, 7, and 9 cycles). The results showed that while the ratio of MM-genet to MM-meth increased slightly with additional index PCR cycles, the difference was not statistically significant (see Figure A below). This is likely due to the optimized KAPA HiFi Uracil+ reagent used in index PCR, which maintains relatively uniform amplification efficiency across fragments with varying GC content.

Additionally, we assessed the impact of input DNA amounts by preparing libraries with 2 ng, 10 ng, and 30 ng of cfDNA, all processed using three cycles of complementary strand synthesis and nine cycles of index PCR. The resulting MM-genet to MM-meth ratios remained consistent across different input amounts (see Figure B below), indicating that varying input amounts do not necessitate adjustments to the number of complementary

strand synthesis cycles.

6. In the target capture experiments described, the number of reads captured from MM-meth is an order of magnitude lower than from MM-genet. Can the authors explain this striking difference?

Response: We apologize for the confusion caused by the insufficient description here.

The significant difference in the number of reads between MM-meth and MM-genet is primarily due to the three cycles of complementary strand synthesis employed in our experiment. When MM-seq is used to detect low-frequency somatic mutations, such as those in cfDNA samples, we would like to obtain as many as possible unique reads in MM-genet. However, the mutation-protective strands that are newly synthesized can be partially damaged during the conversion process, resulting in the loss of templates and a reduced sensitivity in variant detection. To mitigate this issue, we employed a strategy that involved increasing the number of pre-amplification cycles to compensate for the template loss during conversion, which could lead to a significantly higher number of reads from MM-genet compared to MM-meth. We selected three cycles of pre-amplification for our final experiment based on our testing with different cycle numbers (**Appendix Fig.S4B**).

We have now included this clarification in the revised manuscript (lines 211-217). Additionally, the efficiency difference between methylation probes and genetic sequence probes may also contribute to this discrepancy.

7. The discussion is comprehensive but needs to be toned down. The authors are affiliated to a commercial company, and in its current state, the discussion feels somewhat overblown and in part a marketing exercise.

Response: We thank the reviewer for bringing this concern to our attention. While we are affiliated with a commercial company, our discussion is strictly intended to provide a comprehensive scientific analysis rather than serve as a promotional statement. The discussion is grounded in empirical data from our study and supported by existing literature. It has been well established that assessing multiple liquid biopsy biomarkers can enhance the accuracy of MRD detection, and ctDNA analysis benefits from integrating both genetic and epigenetic signatures (PMID: 39609625, 33926918, 37452374). Additionally, the patient's immune status may influence whether (and when) disease progresses, and methylation profiling through liquid biopsy can provide insights into immune dynamics (PMID: 39609625, 34842142). These reports highlighted the clinical need for simultaneous mutation and methylation analysis, and several co-detection methods have been reported in succession. Our study aimed to introduce a more clinically adaptable co-detection approach, and the discussion was intended to highlight the broader prospects of co-detection in clinical diagnosis. We hope our clarifications help address any concerns regarding its tone and intent.

8. Can the authors clarify how duplication rates were calculated? The use of UMIs in this context is unclear from the methods section.

Response: We apologize for the unclear description of data processing in the original manuscript. As the standard EM-seq protocol does not incorporate UMIs, we used GATK MarkDuplicates to calculate duplication rates for methylation analysis to ensure comparability between EM-seq and MM-seq. For mutation analysis, we defined duplicates as reads with aligned genomic coordinates within 5 base pairs and UMI sequences that had no more than 1 mismatch. This process has now been clarified in the data processing section of the Methods (lines 605-608).

Referee #2 (Remarks for Author):

This manuscript has demonstrated a new method called MM-seq to profile the genetic and methylation information from DNA by introducing modified dCTP protective from base conversion. Furthermore, the authors have shown potential clinical application of this method for cancer detection and monitoring based on targeted sequencing or qPCR. The main novelty of the technique is the simultaneous detection of mutations and methylation from the same DNA molecule while being compatible with very low quantity of samples.

The detailed concerns are listed as follows:

We appreciate the reviewer's thorough evaluation of our manuscript and recognition of the novelty and potential clinical application of MM-seq. We thank the reviewer for the valuable feedback, which has helped us to refine and further strengthen our work. We have carefully considered the detailed concerns and responded to each as follows.

Major points

1. What's the proportion of MM-meth and MM-genet reads? After 3 cycles of complementary strand synthesis for both Watson and Crick strands with modified dCTP amplification (line 98-99), the proportion of template remained for MM-meth and MM-genet profiling should be around 1:7 if assuming 100% PCR amplification efficiency. Will the difference lead to sequencing imbalance especially for methylation profiling? For genomic DNA, reads from MM-meth is around 2/3 of that from MM-genet (line 142-144). However, in the Method section, the pre-amplification cycles is 1 for gDNA and 5 for cfDNA (line 467-468). Please elaborate the effect of amplification cycles on the sequencing imbalance, especially for low input samples, such as cfDNA?

Response: We appreciate the reviewer's insightful question regarding the proportion of MM-meth and MM-genet reads, and we apologize for the misstatement of "5 cycles for cfDNA" instead of "3 cycles for cfDNA" (this has been corrected in the revised manuscript, **line 529**).

Theoretically, the proportion of MM-meth and MM-genet reads changes as the number of complementary strand synthesis cycles increases. To investigate this effect, we had conducted an experiment using 30 ng cfDNA samples from healthy donors, testing

different numbers of complementary strand synthesis cycles (1, 3, and 5), each in triplicate. The results showed that increasing the number of pre-amplification cycles led to a rise in MM-genet reads, though not in a strictly linear manner due to amplification inefficiencies (**lines 211-217, Appendix Fig.S4B**). This could be attributed to the nature of the templates after the first round of amplification, which may form hybrid strands of ordinary DNA and dpyC strands or double strands of pure 5py-dC DNA, both of which exhibit reduced amplification efficiency.

We used different pre-amplification cycles for gDNA and cfDNA because we want to maximize unique MM-genet reads for detecting low-frequency somatic mutations in cfDNA samples. Since increasing pre-amplification cycles from three to five did not significantly enhance MM-genet reads, we selected three cycles for our cfDNA experiments.

Although additional pre-amplification cycles alter the proportion of templates available for MM-meth and MM-genet profiling, they do not lead to sequencing imbalance, even for low-input cfDNA samples. This is supported by our results showing that both MM-meth and MM-genet achieved sufficient coverage for methylation and genomic alteration profiling with 15 ng of cfDNA (**lines 222-226, Figure 3A**). Additionally, we also evaluated methylation concordance using 10 ng of cfDNA from three healthy donors, where MM-seq demonstrated a high correlation with standard EM-seq at an average depth of 892× (**lines 239-241, Figure 3C**), further confirming that the amplification cycles do not affect methylation profiling, even for low-input cfDNA samples.

2. Is the high C to T, G to A rate in MM-seq compared to WGS mainly from modified base conversion (line 167-168 and Figure EV3B)? Even though the overall rate of C to T mutations is low, but it is substantially higher than WGS. Exclusion of C to T and G to A mutation profiling (Method Session, line 529-530) could largely limit the application of the MM-seq method.

Response: We agree with the reviewer on the limitation of current MM-seq regarding the high C to T and G to A transition rate. The elevated rate of these transitions primarily stems from modified base conversion, as evidenced by the varying C to T rates observed when using cytosine analogs with differing levels of APOBEC resistance (lines 147-149, Appendix Fig.S1B). While we have successfully reduced this rate to a low level using dpyCTP in pre-amplification, it remains higher than that observed in WGS, posing challenges in identifying C to T and G to A mutations. To mitigate this issue, one potential approach is to leverage information from the paired primary strand in MM-meth, where guanine remains unaltered during conversion (lines 464-468 in the Discussion section). However, we acknowledge that this strategy is only viable for ultra-deep sequencing, where sufficient paired MM-genet and MM-meth reads are available. Recognizing this limitation, we are actively exploring alternative cytosine analogs to further refine MM-seq and enhance its accuracy in future iterations.

3. The authors should also generate a mutational spectrum to compare the different types of mutations from MM-seq and WGS more clearly.

Response: We appreciate the reviewer's thoughtful suggestion. In response, we have

now included a comprehensive mutational spectrum comparing all twelve types of base transitions between MM-seq and WGS (Appendix Fig.S3B). This provides a clearer comparison of the mutation types detected by both methods.

4. The authors have reported the duplication rate of MM-meth and MM-genet from target sequencing with 15ng of cfDNA input (line 196-199). Clinical available cfDNA could be as low as several nanogram or even at sub-nanogram level. It's important to evaluate the sequencing parameters, such as duplication level and unique sequencing depth with lower cfDNA input since the authors have prepared library with 2ng and 10ng of cfDNA as input (line 185-187).

Response: We agree with the reviewer that evaluating sequencing parameters at lower cfDNA input levels is crucial for clinical applications. In the previous version of the manuscript, only one cycle of complementary strand synthesis was performed during library preparation for 2 ng and 10 ng cfDNA samples, as the primary goal was to verify MM-seq's capability in handling low-input DNA. However, the products generated from a single pre-amplification cycle were insufficient for target capture, preventing us from assessing sequencing parameters with these libraries. To address this concern, we have reproduced the libraries using 2 ng and 10 ng of a mixed cfDNA sample with three cycles of pre-amplification, each in triplicate. We assessed the duplication rate and unique sequencing depth using approximately 2.1 million MM-meth paired-end reads and 16.5 million MM-genet paired-end reads per sample. The results demonstrate that MM-seq can achieve coverage exceeding 300× for both MM-meth and MM-genet, even with as little as

2 ng of DNA (see figure below). This level of coverage is sufficient for methylation and genetic alteration profiling in certain clinical scenarios. For example, a mean coverage of 90.87 \times has been reported as adequate for mutation detection (PMID: 37373553), and a mean coverage of 139 \times has been shown to support reliable methylation profiling (PMID: 33506766).

5. How much cfDNA has been obtained from the 10 NSCLC patients for comparison of MM-seq and standard DNA-seq (line 202-205)?

Response: We apologize for not providing this information earlier. The total cfDNA obtained from each of the 10 NSCLC patients, as well as the amount of cfDNA input used for both MM-seq and standard DNA-seq, is now included in lines 228-232 and Table EV1 of the revised manuscript.

6. Are the sequencing depth and duplication level comparable between MM-seq and standard DNA-seq for MAF profiling (line 205-207)?

Response: We appreciate the reviewer's insightful question. The sequencing depth and

duplication level were not comparable between MM-seq and standard DNA-seq due to differences in cfDNA input. A fixed input of cfDNA (30 ng) was used for standard DNA-seq as a benchmark, leading to cfDNA available for MM-seq varied between 3.68 and 29.75 ng per sample. Due to this discrepancy in cfDNA input, the sequencing depth and duplication level were not directly comparable between the two methods (**Table EV1** in the revised manuscript). Nevertheless, despite the lower cfDNA input in MM-seq, all mutations were successfully detected, and MM-seq demonstrated a strong correlation in MAF with standard DNA-seq, underscoring its reliability in mutation detection.

7. The authors have demonstrated the ability of MM-seq to detect allele-specific methylation level with targeted profiling for the selected 8 CpGs around 5 SNPs (line 224-225)? Are there any false positives if not targeting known ASM regions?

Response: We understand the reviewer's concern regarding potential false positives in allele-specific methylation (ASM) detection using MM-seq. Our study focused on identifying true positive ASM events, as we currently lack reference data to definitively classify regions outside well-characterized ASM loci as true negatives or potential ASM sites. As a result, we did not specifically evaluate regions without previously documented ASM. However, in response to this comment, we examined all CpG sites within our target region that could not be conclusively classified as ASM or non-ASM based on reference data. The results showed that while some CpG sites exhibited smaller methylation differences between the reference and alternative alleles compared to known ASM sites, these differences remained statistically significant (p -value < 0.001 , see figure below). In

future studies, we plan to extend our analysis to broader or genome-wide assessments, including the identification of true negative-control ASM regions, to further evaluate the specificity of MM-seq across diverse genomic contexts.

8. The superior performance for NSCLC detection of MM-seq compared to DNA-seq seems to be completely contributed by methylation profiling (line 269-273, Figure 4C). Using mutations it seems MM-seq was totally inadequate in detecting NSCLC samples. As such it is not meaningful to compare MM-seq with targeted DNA-seq for cancer detection. A comparison should be made between MM-seq and targeted methylation sequencing for cancer detection.

Response: We recognize the reviewer's concerns regarding the evaluation of MM-seq for ctDNA detection. It is true that the improved performance of MM-seq in ctDNA detection is primarily attributed to methylation profiling. The comparison with standard DNA-seq, rather than targeted methylation sequencing, was made because mutation detection is essential in routine cancer clinical diagnostics, particularly for guiding targeted therapies. However, in certain clinical scenarios, such as MRD monitoring, companion diagnostics based solely on mutation detection may not be sufficient, as MRD detection requires high

sensitivity for ctDNA detection. MM-seq was compared to standard DNA-seq to demonstrate that, by integrating methylation profiling, it enhances ctDNA detection while still maintaining the ability to identify mutations for companion diagnostics.

Although we did not conduct a direct comparison between MM-seq and targeted methylation sequencing in ctDNA detection, we have compared the methylation assessment between MM-seq and EM-seq. The depth was comparable and CpG methylation was highly correlated between the two techniques (**lines 176-178, Figure 2C; lines 239-241, Figure 3C**). We have also compared MM-seq with a study using ELSA-seq for ultrasensitive detection of circulating tumor DNA through deep methylation sequencing (PMID 34131323). At a similar level of PCR duplicates (~70%), MM-seq achieved a mean unique on-target depth of 1,722× for epigenetic reads with 15 ng of cfDNA, whereas ELSA-seq achieved 615× and 631× unique depth for 10 ng and 30 ng cfDNA, respectively (see table below), indicating a possible improved performance in methylation profiling for MM-seq.

	MM-seq (N = 13)	ELSA-seq (N = 2)	ELSA-seq (N = 2)
Data source	In-house	ELSA-seq study	ELSA-seq study
DNA input	15 ng	10 ng	30 ng
duplicate rate	68.7% (64.2% - 71.6%)	72.4% (72 % - 72.8%)	70.9% (70.6% - 71.1%)
unique depth	1,722 (1,036 - 2,115)	615 (697 - 633)	631 (620 - 642)

9. The results from qPCR profiling of low-level SDC2 hypermethylation seems to be not

highly repeatable for both MM-qPCR and standard qPCR (Figure 6). The authors should repeat at least 3 times for each qPCR samples for both mutation and methylation profiling. It's necessary to provide results from more repeats to prove the performance consistency. On the other hand, the authors should also include performance comparison of MM-qPCR with other commercial kits that measure KRAS G13D mutation and SDC2 methylation, instead of only with their own kits.

Response: We appreciate the reviewer's valuable feedback regarding the need for additional replicates in the qPCR testing. In response to this suggestion, we have now performed three repetitions for each qPCR sample, testing dilutions ranging from 0.1% to 3%, and have included the results for these three replicates in the revised manuscript (see **lines 371-383 and Figure 6**). Although MM-qPCR was not able to detect *KRAS* G13D mutation and *SDC2* methylation at 0.1% in all three duplicates, it consistently detected both *KRAS* G13D mutation and *SDC2* methylation at 0.5% in all three duplicates. This performance matches the limits of detection for both the KRAS Mutation Detection Kit (1%) and the SDC2 Methylation Detection Kit (1.25%), like standard qPCR does. As this work primarily focused on validating the adaptability of MM co-detection across different platforms using a prototype, a direct performance comparison with other commercial kits was not included at this stage. However, we are actively optimizing the MM-qPCR protocol, and we plan to conduct comparisons with other commercial kits in future studies.

10. Since a major goal of the method is for clinical applications, more discussion should be made in terms of the additional time/cost, in terms of laboratory work and

bioinformatics, that may be required to implement MM-seq in a clinical lab.

Response: We appreciate the reviewer's suggestion for additional discussion on the practical implications of implementing MM-seq in a clinical setting. Currently, our method remains in a proof-of-concept stage and has not yet been fully optimized or validated for routine clinical use. Therefore, immediate clinical implementation, particularly across diverse hospital or reference-lab environments, is not yet feasible. Nonetheless, to address the reviewer's concern, we have expanded our manuscript, including preliminary workflow and cost considerations, potential barriers to clinical adoption and future directions for streamlining (lines 406-410, lines 476-482, Appendix Table S5). While MM-seq is not yet ready for immediate clinical deployment, we believe that detailing these considerations, along with our road map for further optimization, provides a clearer understanding of the practical steps needed before MM-seq can be reliably implemented in a clinical laboratory. We hope this additional discussion will help readers appreciate both the potential of our method and the current limitations that must be addressed in future development.

Other minor points:

11. Line153 should be describing Fig 2A.

Response: We apologize for the incorrect figure reference. The text has been corrected to refer to Fig 2A in the revised script (line 171).

12. Typo in line 293 with extra "s": "formalin-fixed paraffin-embedded (FFPE) samples

from ovarian and breast cancer patients s"

Response: We apologize for the typo. It has been corrected in the revised manuscript (**line 327**).

13. Line 332-333 should be short turnaround time, not "short turnaround times"

Response: We apologize for the miswording. The text has been corrected to "short turnaround time" in the revised manuscript (**lines 366-367**).

Referee #3 (Comments on Novelty/Model System for Author):

Technical Quality (including statistical analysis): Medium

The technical quality of the study is generally good; however, there are notable concerns that warrant further attention. These include the potential for false-positive mutation detection, limited sample sizes, and the lack of replication in key experiments. Thus, while the methodology demonstrates promise, further rigorous testing is required to ensure its robustness.

Novelty: High

The study presents a highly novel approach, specifically the MM-seq method for ctDNA

detection, which offers a significant advancement in diagnostic sensitivity and has the potential to enhance therapeutic decision-making. This innovation could provide a valuable contribution to the field, though its clinical applicability requires further validation.

Medical Impact: Medium

The potential medical impact of the study is considerable, as the proposed methodology could improve the detection and monitoring of ctDNA in clinical settings. However, the current study's limited sample size and the absence of validation with tumor tissue sequencing reduce the immediate applicability of the findings to clinical practice. Further studies with larger cohorts are needed to fully assess the clinical utility of the approach.

Adequacy of Model System: Adequate

The model system employed in this study is adequate in demonstrating the feasibility of MM-seq for ctDNA analysis. However, several limitations need to be addressed to improve its clinical relevance. These include concerns regarding the accuracy of mutation detection. Further validation with known mutations and larger clinical cohorts would significantly enhance the robustness of the results shown.

Referee #3 (Remarks for Author):

The reviewed article, "A DNA alteration and methylation co-detection method for clinical purpose," addresses a critical gap in current genomic and epigenomic diagnostics.

Traditional methodologies often require separate workflows for DNA mutation and methylation detection, which can be resource-intensive and impractical, particularly in clinical contexts with limited sample availability.

The authors propose a novel co-detection method (MM-seq) designed to overcome these limitations by integrating the detection of mutations and methylation into a streamlined workflow. By utilizing a mutation-protective strand synthesis with modified deoxycytidine triphosphates (dCTPs), this approach offers enhanced sensitivity, specificity, and adaptability to various diagnostic platforms, including next-generation sequencing (NGS) and qPCR.

The article is well-written and presents an interesting exploration of a clinically relevant topic. While the detailed methodology and comprehensive analysis highlight the innovation's potential to improve diagnostic efficiency and sensitivity in circulating tumor DNA (ctDNA) detection, further experiments are needed to rule out false positive mutation detection. Additionally, more robust results are required to demonstrate the extrapolation of these findings to clinical settings and their impact on therapeutic decision-making.

Specific points where the article should be revised to meet the standards of a high-impact journal such as EMBO Molecular Medicine will be outlined in the following sections.

We appreciate the reviewer's recognition of the clinical relevance and methodological innovation in our co-detection strategy, MM-seq. Thank the reviewer for the thorough and constructive feedback, which has helped us significantly strengthen our manuscript.

Below we address each of the concerns raised:

MAJOR POINTS

INTRODUCTION

Page 2, lines 67-68:

- The authors state, "Furthermore, most existing methods lack sufficient clinical validation to confirm their efficacy in diagnostics." This is a significant claim and should be substantiated with references to relevant studies or data.

Response: We understand the reviewer's concerns regarding the statement on the clinical validation for existing co-detection methods. In the original publications describing four representative techniques for simultaneous methylation and mutation detection: TAPS, Methyl-SNP-seq, 5-letter seq, and MethylSaferSeqS, only MethylSaferSeqS has been evaluated for mutation detection in cfDNA samples from 19 patients with advanced colorectal cancer. In response to this comment, we further reviewed studies related to these techniques. As of March 1, 2025, these four methods have been cited 154, 1, 27, and 7 times, respectively. However, except for TAPS, which has been used in a diagnostic accuracy study in Jan, 2025 (PMID 39779727), none of the other methods have been applied in real-world clinical samples. Still, TAPS has limitations: it cannot detect mutations at CpG sites where methylation is also called, and it may encounter challenges in detecting mutations within regions spanning CpG sites when used in qPCR-based platforms. We have incorporated this information into the revised manuscript to clarify our statement regarding the current state of clinical validation for these methods (line 83-85).

PRINCIPLE OF MM-SEQ

Page 4, line 126:

- The authors note that "Libraries using dpyCTP demonstrated the highest proportion of perfectly matched UMIs (95.45%, Supplementary Fig EV1A)." This percentage seems not optimum for ultrasensitive mutation tests for MRD detection. Concerns arise regarding whether the method introduces mutations in UMIs or in the inserts during bisulfite conversion. This issue is not tested with the whole-genome sequencing experiments provided. The authors should perform whole-exome sequencing or gene panel sequencing at optimal depth using DNA with known mutations to evaluate their method's accuracy and the rate of false-positive mutations introduced.

Response: We understand the reviewer's concern regarding the possibility of mutations introduced during the bisulfite conversion process. In our current implementation of MM-seq, false C to T (or G to A) transitions may occur due to partial deamination of pyC. We have evaluated the error rates of all mutation types in our whole-genome sequencing experiments (**Appendix Figure S3B**). To mitigate this, we have excluded C to T (G to A) mutations when performing deep gene panel sequencing (described in the Method section, **line 595-596**) and have proposed an error-correction strategy that utilizes paired reads between MM-meth and MM-genet (**line 464-468**). Moreover, we are actively screening cytosine analogs with enhanced resistance to APOBEC deamination to further reduce these errors. We have now discussed these limitations and outlined our future directions in the revised manuscript's Discussion.

ASSESSMENT OF APPLYING MM-SEQ IN CLINICAL PRACTICE

Page 5, lines 192-196:

- The authors mention, "These libraries were captured using a hybrid panel designed for ctDNA assessment, incorporating methylation probes targeting approximately 21 kb of the genome and sequence probes targeting approximately 38 kb (detailed in the methods section under ctDNA detection). MM-seq generated a mean of 2.1 million paired-end reads for MM-meth and 20.2 million paired-end reads for MM-genet..." It is unclear why such a significant difference in paired-end reads exists between methylation and sequence probes if the targeted regions are of similar size. Clarification or additional data addressing this discrepancy is needed.

Response: We apologize for the confusion caused by the lack of critical information. The significant difference in paired-end reads between MM-meth and MM-genet is primarily due to the three cycles of pre-amplification used in the library preparation. When MM-seq is used to detect low-frequency somatic mutations, such as those in cfDNA samples, we would like to obtain as many as possible unique reads in MM-genet. However, the mutation-protective strands that are newly synthesized can be partially damaged during the conversion process, resulting in the loss of templates and a reduced sensitivity in variant detection. To mitigate this issue, we employed a strategy that involved increasing the number of pre-amplification cycles to compensate for the template loss during conversion, which could lead to a significantly higher number of reads from MM-genet compared to MM-meth. We selected three cycles of pre-amplification for our final

experiment based on our testing with different cycle numbers (Appendix Fig.S4B). The details of this critical step have now been clarified in the revised manuscript (lines 211-217). Additionally, the difference in efficiency between the methylation probes and genetic sequence probes may also contribute to the imbalance in read numbers between MM-meth and MM-genet.

Figure 3B:

- The specific mutant allele frequency (MAF) values should be displayed.

Response: We appreciate the reviewer's suggestion. As suggested, we have added the specific MAF values to Figure 3B to and included the corresponding data in Table EV1.

Page 6, lines 228-230:

- "By pairing reads from MM-genet and MM-meth based on mapping positions and UMI sequences, 36.28% of fragments in MM-meth found their pairs in MM-genet and were allocated to discrete alleles of the target SNPs." This percentage of paired meth and genet reads seems low for reliable cis methylation and genetic alteration detection in MRD protocols. This limitation should be explicitly acknowledged and discussed.

Response: We acknowledge that the current percentage of paired meth and genet reads in MM-seq may pose challenges for reliably detecting cis methylation and genetic alterations at low frequencies. At present, we primarily use it to detect SNP allele-specific methylation, which requires fewer paired reads. However, we are actively working to improve DNA fragment recovery for MM-seq and aim to increase the proportion of paired

meth and genet reads through enhancements in fragment recovery and by optimizing the number of pre-amplification cycles. We have discussed these limitations and our planned improvements in the Discussion section of the revised manuscript (**lines 468-472**).

MM-SEQ ENHANCES TUMOR DETECTION AND MONITORING

Page 7, lines 264-268:

- The authors report a higher ctDNA-positive detection rate with MM-seq (26.92%) compared to traditional methods (15.38%). However, they should clarify the criteria used to distinguish tumor-derived mutations from artifacts or false positives. Tumor tissue sequencing should be included to confirm the somatic nature of detected mutations.

Response: We agree with the reviewer that confirming somatic mutations is crucial for ensuring the reliability of ctDNA detection. To distinguish tumor-derived mutations from false positives, we used the number of supporting reads, variant allele frequency, and paired whole blood cell sequencing. We have revised the Methods section to clarify these criteria in more detail. (**lines 591-595**). We also acknowledge that including tumor tissue sequencing would further strengthen the validation of mutation calls. However, not all patients had paired tumor tissue samples available in this study. Of the four samples in which somatic mutations were detected, three had paired tumor tissue available. We performed standard DNA-seq on these three tumor tissue samples and observed that the mutations identified in the corresponding cfDNA samples were indeed present in the paired tumor samples (see table below), confirming the tumor-derived nature of the mutations.

Sample	Mutation	Tissue availability	VAF in cfDNA	VAF in tumor tissue
Sample 1	TP53 (c.415A>T)	Available	0.40%	52.74%
Sample 2	KRAS (c.182A>C)	Not available	0.23%	/
Sample 3	TP53 (c.722C>G)	Available	1.07%	5.37%
Sample 4	EGFR (c.2573T>G)	Available	0.13%	63.56%

- The clinical sample size (26 NSCLC patients) is insufficient for robust conclusions. At least 50 cfDNA samples should be analyzed to strengthen the validity of the study.

Response: We fully acknowledge that 26 samples may be insufficient for drawing broad clinical conclusions. However, our study was intended as an initial feasibility assessment for using a MM-seq prototype to enhance ctDNA detection, rather than large-scale clinical validation. While the current cohort is limited, our results provide preliminary yet encouraging evidence supporting the feasibility of this approach. We plan to conduct future studies with larger and more diverse cohorts to validate and strengthen these findings. This limitation and future options now have been discussed in the revised manuscript (lines 473-482).

Page 7, lines 295-296:

- The authors categorize samples into HRD (N=16) and HRP (N=11) groups. The sample size is too small to draw meaningful conclusions. Both groups should include at least 30 samples.

Response: We agree that the sample sizes in both the HRD (N=16) and HRP (N=11) groups are insufficient for drawing clinically meaningful conclusions. However, the primary goal of this study was to explore the feasibility of using MM-seq to assess the relationship between changes in gene function and genomic status, rather than to provide definitive conclusions. The panel is conceptual, and the assay is still in its prototype phase. While we acknowledge the small sample sizes, this proof-of-concept study has provided valuable initial insights. This limitation has been highlighted in the Discussion section (lines 473-482), and we plan to optimize our methodology and expand the sample sizes in future studies to strengthen the statistical power and further validate our observations.

- Clinical characteristics of the patients should be presented in table/s. It is crucial to specify whether the breast cancer patients included are early-stage or metastatic. The methods section should confirm adherence to ethical standards.

Response: We appreciate the reviewer's suggestion to include clinical characteristics. This information, including tumor stage for breast cancer patients, is now provided in **Table EV2**. Additionally, adherence to ethical standards has been confirmed in the Methods section (lines 500-502).

MM-QPCR VALIDATION

Page 9, lines 349-353:

- The authors compare MM-qPCR results with standard methods but only perform the experiments once. Replicates should be performed in triplicate, and additional dilutions between 1% and below 0.1% should be explored. Testing with real plasma samples from tumor patients is recommendable to assess the methodology's practical efficiency.

Response: We thank the reviewer for the suggestion to include additional replicates and dilutions in the assessment of MM-qPCR. As suggested, we repeated the experiments with three additional dilutions (0.05%, 0.5%, and 3%) and performed triplicate tests for each dilution (lines 371-383). Although MM-qPCR was unable to detect the *KRAS* G13D mutation and *SDC2* methylation at 0.1%, the lower dilution reported in the previous manuscript version, it reliably detected both *KRAS* mutation and *SDC2* methylation at 0.5% across all three replicates (Figure 6). This performance reached the detection limits for the two kits used (1% for the *KRAS* Mutation Detection Kit and 1.25% for the *SDC2* Methylation Detection Kit). As our primary goal was to demonstrate the cross-platform adaptability of the MM co-detection using an assay prototype, we have not yet assessed its performance with real tumor patient plasma samples. However, this will be part of future validation studies once the assay is optimized.

29th Apr 2025

Dear Dr. Yang,

Thank you for submitting your manuscript to EMBO Molecular Medicine. We have now received the reports from the initial referees.

As you will see below, while referee #2 is satisfied with the revisions, referees #1 and #3 still have a few concerns. We discussed these reports within the team, and we agreed that while additional clinical validation will NOT be required for further consideration of the manuscript in EMBO Molecular Medicine (although welcome), careful rewriting of the manuscript to address the concerns of both reviewers will be mandatory.

As EMBO Press usually encourages one single round of revisions, please be aware that this will be the last chance for you to address the referees' concerns. The revised manuscript will once again be subjected to review, and we cannot guarantee a positive outcome at this stage.

Moreover, please address the following editorial requests:

1. Emails to yuanwz@amoydx.com and jiangqing.wang@amoydx.com bounced, please check and correct.
 2. Please note that all corresponding authors are required to supply an ORCID ID for their name upon submission of a revised manuscript. An ORCID ID is currently missing for Frank Ron Zheng.
 3. The Data Availability Section should be moved after Methods. Please provide the specific URL for PRJCA030061.
 4. The funding information provided in the Acknowledgements section should match the funding information provided in the submission system.
 5. Please rename the conflict of interests section "Disclosure statement and competing interests". We updated our journal's competing interests policy in January 2022 and request authors to consider both actual and perceived competing interests.
 6. Please correct the headings/titles of the EV tables to "Table EV1" and "Table EV2" instead of datasets.
 7. We replaced Supplementary Information with Expanded View (EV) Figures and Tables that are collapsible/expandable online. EV Figures should be cited as 'Figure EV1, Figure EV2" etc... in the text and their respective legends should be included in the main text after the legends of regular figures.
 - For the figures that you do NOT wish to display as Expanded View figures, they should be bundled together with their legends in a single PDF file called *Appendix*, which should start with a short Table of Content. Appendix figures should be referred to in the main text as: "Appendix Figure S1, Appendix Figure S2" etc.
 - Additional Tables/Datasets should be labeled and referred to as Table EV1, Dataset EV1, etc. Legends have to be provided in a separate tab in case of .xls files. Alternatively, the legend can be supplied as a separate text file (README) and zipped together with the Table/Dataset file.
- See detailed instructions here:
<https://www.embopress.org/page/journal/17574684/authorguide#expandedview>
8. Thank you for providing Source Data. Please upload them as one file per figure.
 9. Every published paper now includes a 'Synopsis' to further enhance discoverability. Synopses are displayed on the journal webpage and are freely accessible to all readers. They include a short stand first (maximum of 300 characters, including space) as well as 2-5 one-sentences bullet points that summarizes the paper. Please write the bullet points to summarize the key NEW findings. They should be designed to be complementary to the abstract - i.e. not repeat the same text. We encourage inclusion of key acronyms and quantitative information (maximum of 30 words / bullet point). Please use the passive voice. Please attach these in a separate file or send them by email, we will incorporate them accordingly.
 10. Please also suggest a visual abstract to illustrate your article as a PNG file 550 px wide x 300-600 px high. A cropped portion of this image will serve as thumbnail for the table of content on our webpage.
 11. Please address the queries from our copy editors in the figure legends:
 - Please note that the exact p values are not provided in the legends of figures 3D, 4D, 5C, S5B
 - Please indicate the statistical test used for data analysis in the legends of figures 3A, 4D, 5C; S4 C, S5B
 - Please note that the box plots need to be defined in terms of minima, maxima, centre, bounds of box and whiskers, and percentile in the legends of figures 4D, 5C, S5 B
 - Please note that information related to n is missing in the legends of figures 3A, 4D, 5C, S3B, S4 B, S5 B
 - Please note that n=2 in figures S2A, C; S3 A
 - Please note that the error bars are not defined in the legends of figures 3A, S1A-C; S3 B; S4 B

As part of the EMBO Publications transparent editorial process initiative (see our Editorial at <http://embomolmed.embopress.org/content/2/9/329>), EMBO Molecular Medicine will publish online a Review Process File (RPF) to accompany accepted manuscripts.

In the event of acceptance, this file will be published in conjunction with your paper and will include the anonymous referee

reports, your point-by-point response and all pertinent correspondence relating to the manuscript. Let us know whether you agree with the publication of the RPF and as here, if you want to remove or not any figures from it prior to publication. Please note that the Authors checklist will be published at the end of the RPF.

I look forward to receiving your revised manuscript.

Sincerely,

Lise Roth

**** Reviewer's comments ****

Referee #1 (Remarks for Author):

In this revised manuscript, the authors provide a clear rebuttal to the comments I made earlier. I particularly appreciate the additional work done by the authors, and the clarifications provided in the manuscript. I have no additional questions.

I do however disagree with your reply that the discussion does not need to be toned down: phrases such as "MM-seq was tailor-made for plug-and-play application" or "its versatility allows it to be adapted to various platforms and integrated seamlessly with commercially available kits" are examples of language that I know from marketing brochures, not from a scientific document.

As a final comment, I would have preferred to see my comment on nucleotide purity addressed in the manuscript, not simply in the rebuttal.

Referee #2 (Comments on Novelty/Model System for Author):

There is still remain some gap for clinical application, however the manuscript is sufficient to demonstrate a proof of concept and future potential.

Referee #2 (Remarks for Author):

Thank you for addressing queries. I have no further concerns for publication.

Referee #3 (Comments on Novelty/Model System for Author):

1. Technical Quality: High

The technical development of the MM-seq method is carefully designed and nicely implemented. The authors demonstrate convincing analytical validation, including comparisons with established techniques (EM-seq, WGS) and thoughtful optimization of enzymatic conversion and UMI-based read assignment. The statistical analysis appears appropriate for the scope of the work.

2. Novelty: Medium

While the combination of methylation and mutation detection in a single assay is not entirely novel, the methodological implementation here - particularly the use of modified dCTPs and the dual read strategy - offers a streamlined workflow that improves upon prior approaches. Nonetheless, the conceptual advance is incremental rather than transformative.

3. Medical Impact: Low

Despite the technical merit, the current clinical validation is insufficient to support the claimed applications, especially in the context of minimal residual disease (MRD) detection. The limited number of clinical samples, combined with low ctDNA detection rates even at baseline, restricts the translational impact of the study. Additional patient cohorts and longitudinal data are needed to substantiate clinical relevance.

4. Adequacy of Model System: N/A

Referee #3 (Remarks for Author):

While the method is well developed and appears to perform reliably in analytical evaluations, the claims regarding its clinical applicability-particularly for minimal residual disease (MRD) detection-are not adequately supported by the data presented. The authors place considerable emphasis on MRD detection, yet in their own data, ctDNA is not detectable in the majority of baseline samples from NSCLC patients, even before treatment. This is particularly problematic, as baseline samples typically exhibit higher ctDNA levels than genuine MRD settings. If MM-seq fails to detect ctDNA reliably in these pre-treatment samples, it is unlikely to succeed in the post-treatment MRD context, where VAFs may fall below 100 parts per million. To support the claimed clinical utility of MM-seq for MRD detection, it is imperative that the authors expand their clinical validation to include a larger number of patient samples-ideally covering both early-stage disease and post-treatment timepoints-and demonstrate robust detection of ctDNA at very low VAFs. Without such evidence, the assertions regarding MRD detection are premature.

Therefore, I strongly recommend that the authors revise the manuscript by:

1. Substantially toning down the language in the abstract, results, and discussion regarding the utility of MM-seq for MRD detection.
2. Explicitly acknowledging the limited sensitivity shown in the current cohort, and the need for further validation in true low-ctDNA clinical contexts.
3. Expanding the number of clinical samples analyzed, particularly from patients with early-stage disease or post-treatment status, to properly assess the method's utility for ctDNA detection at low VAFs.

While MM-seq is a promising technology, the current data do not justify the claims made. I would support publication only after a careful revision of the manuscript's tone and conclusions, or ideally, after additional clinical validation is included.

**** Reviewer's comments ****

Referee #1 (Remarks for Author):

In this revised manuscript, the authors provide a clear rebuttal to the comments I made earlier. I particularly appreciate the additional work done by the authors, and the clarifications provided in the manuscript. I have no additional questions.

I do however disagree with your reply that the discussion does not need to be toned down: phrases such as "MM-seq was tailor-made for plug-and-play application" or "its versatility allows it to be adapted to various platforms and integrated seamlessly with commercially available kits" are examples of language that I know from marketing brochures, not from a scientific document.

As a final comment, I would have preferred to see my comment on nucleotide purity addressed in the manuscript, not simply in the rebuttal.

Response: We thank the reviewer for the thorough evaluation of our revised manuscript and for the positive feedback regarding the clarifications and additional work provided. We also appreciate the reviewer's continued engagement and valuable suggestions, which have contributed to improving the quality of the manuscript. In response to the reviewer's comments:

1. We acknowledge the reviewer's concern regarding the tone of certain statements in the discussion and appreciate this feedback. In the revised manuscript, we have carefully revised the relevant sections (see tracked changes, **lines 369-467**) to ensure the language remains objective and consistent with academic standards.
2. We agree with the reviewer that the information regarding nucleotide purity is important to include in the manuscript. In response, we have now incorporated a description in the Methods section noting the purity of dCTP analogs (**lines 507-514**),

and expanded the discussion to consider potential reasons for the observed high conversion rate of dmC (lines 434-439).

Referee #2 (Comments on Novelty/Model System for Author):

There is still remain some gap for clinical application, however the manuscript is sufficient to demonstrate a proof of concept and future potential.

Referee #2 (Remarks for Author):

Thank you for addressing queries. I have no further concerns for publication.

Response: We thank the reviewer for the time and consideration in reviewing our manuscript. We appreciate their positive feedback and are pleased that all concerns have been satisfactorily addressed.

Referee #3 (Comments on Novelty/Model System for Author):

1. Technical Quality: High

The technical development of the MM-seq method is carefully designed and nicely implemented. The authors demonstrate convincing analytical validation, including comparisons with established techniques (EM-seq, WGS) and thoughtful optimization of

enzymatic conversion and UMI-based read assignment. The statistical analysis appears appropriate for the scope of the work.

2. Novelty: Medium

While the combination of methylation and mutation detection in a single assay is not entirely novel, the methodological implementation here - particularly the use of modified dCTPs and the dual read strategy - offers a streamlined workflow that improves upon prior approaches. Nonetheless, the conceptual advance is incremental rather than transformative.

3. Medical Impact: Low

Despite the technical merit, the current clinical validation is insufficient to support the claimed applications, especially in the context of minimal residual disease (MRD) detection. The limited number of clinical samples, combined with low ctDNA detection rates even at baseline, restricts the translational impact of the study. Additional patient cohorts and longitudinal data are needed to substantiate clinical relevance.

4. Adequacy of Model System: N/A

Referee #3 (Remarks for Author):

While the method is well developed and appears to perform reliably in analytical evaluations, the claims regarding its clinical applicability-particularly for minimal residual disease (MRD) detection-are not adequately supported by the data presented.

The authors place considerable emphasis on MRD detection, yet in their own data, ctDNA is not detectable in the majority of baseline samples from NSCLC patients, even before treatment. This is particularly problematic, as baseline samples typically exhibit higher ctDNA levels than genuine MRD settings. If MM-seq fails to detect ctDNA reliably in these pre-treatment samples, it is unlikely to succeed in the post-treatment MRD context, where VAFs may fall below 100 parts per million.

To support the claimed clinical utility of MM-seq for MRD detection, it is imperative that the authors expand their clinical validation to include a larger number of patient samples-ideally covering both early-stage disease and post-treatment timepoints-and demonstrate robust detection of ctDNA at very low VAFs. Without such evidence, the assertions regarding MRD detection are premature.

Therefore, I strongly recommend that the authors revise the manuscript by:

1. Substantially toning down the language in the abstract, results, and discussion regarding the utility of MM-seq for MRD detection.
2. Explicitly acknowledging the limited sensitivity shown in the current cohort, and the need for further validation in true low-ctDNA clinical contexts.
3. Expanding the number of clinical samples analyzed, particularly from patients with

early-stage disease or post-treatment status, to properly assess the method's utility for ctDNA detection at low VAFs.

While MM-seq is a promising technology, the current data do not justify the claims made. I would support publication only after a careful revision of the manuscript's tone and conclusions, or ideally, after additional clinical validation is included.

Response: We thank the reviewer for the thoughtful and critical evaluation of our revised manuscript. We appreciate the recognition of MM-seq as an analytically robust technology and fully understand the concerns raised regarding its current clinical utility in the MRD setting. Below, we address each of the reviewer's points in detail:

1. We agree that the original manuscript may have inappropriately overstated the enhancement of MM-seq for MRD detection based on the current level of evidence. We have revised the manuscript to instead describe the relative improvement in ctDNA detection sensitivity, including changes to the abstract (**lines 22-34**), results (**lines 247-287**), and discussion (**lines 380-401, lines 449-460**) sections. The language in these sections has been toned down to present a more measured interpretation. The revised text also emphasizes that while MM-seq shows potential, its utility in low-ctDNA contexts remains to be fully established through further validation.
2. We have revised the discussion to clearly acknowledge the limited sensitivity observed in our cohort and the gap between current sensitivity and reliable ctDNA detection low-ctDNA contexts (**lines 391-393**). Furthermore, we now explicitly note

that additional validation in larger and more appropriate patient cohorts is required to verify if MM-seq can be considered for clinical MRD applications (**lines 450-452**).

3. We fully agree that expanded clinical validation would strengthen the manuscript significantly. However, as this study represents an initial feasibility assessment, and due to constraints in sample availability, we faced challenges in collecting a sufficient number of early-stage or post-treatment samples in the short term. In addition, designing an adequate methylation panel that includes enough informative markers remains a substantial task, requiring considerable effort. Nonetheless, we have a clear plan to conduct future clinical validation to assess performance in low-ctDNA contexts, involving a larger cohort and a well-optimized panel. We have now explicitly acknowledged these limitations and our future plans in the revised discussion (**lines 449-460**).

We thank the reviewer for highlighting the importance of accurate and responsible interpretation of experimental data. The revisions we have made are intended to align the claims in the manuscript with the evidence currently available and to clarify the future directions necessary for clinical translation.

Dear editor Lise,

Thank you for your continued feedback. For the issues that remain to be addressed:

1. The cell lines use in our study were authenticated with STR profiling but were not tested for mycoplasma contamination. This has now been specifically stated in the checklist. Reports on STR profiling are attached.

2. We mistakenly interpreted the checklist item as referring to the origin of the cell lines used in our study. Upon realizing that it pertains to primary cultures directly used in the research, we have corrected the entry to “not applicable” in the checklist.

3. We did not provide a reference number because this study qualifies for exemption from ethical review under Article 32 of the *Regulations on Ethical Review of Life Sciences and Medical Research Involving Humans*, issued by the National Health Commission (NHC) of China (https://www.gov.cn/zhengce/zhengceku/2023-02/28/content_5743658.htm — only available in Chinese). The regulation states:

Life science and medical research involving human information data or biological samples may be exempted from ethical review if it does not cause harm to the human body, does not involve sensitive personal information or commercial interests. This is to reduce unnecessary burdens on researchers and to promote human-related life science and medical research:

1. Research using publicly available data obtained legally, or data generated through observation that does not interfere with public behavior;
2. Research using anonymized information data;
3. Research using existing human biological samples, where the sources comply with relevant laws and ethical principles, the research content and purposes fall within the scope of properly obtained informed consent, and the research does not involve human reproductive cells, embryos, reproductive cloning, chimerism, or heritable genetic modifications;
4. Research using human-derived cell lines or cell stocks obtained from biobanks, where the research content and purposes fall within the scope authorized by the provider, and the research does not involve human embryos, reproductive cloning, chimerism, or heritable genetic modifications.

In our study, DNA samples involving human subjects were sourced from residual samples remaining after the completion of clinical testing in the medical lab. All patients had

provided written informed consent specifying that their residual samples could be used for scientific research. The sample size is small, and all samples were anonymized. No clinical treatment information was collected. Therefore, the study fully complies with the relevant laws, ethical principles, and exemption criteria outlined above, which have been described in the method part of Human samples.

4. We indicated this item in the checklist because data from some, but not all, study participants were collected and disclosed in the manuscript (Table EV2). We have now revised the checklist to include a more specific description in the “In which section is the information available” field.

The revised checklist can be found in the attachment. Please let us know if anything further is required.

Kind regards,

Shuang Yang

R&D Department,

Vice president of Amoydiagnostics,

<https://www.amoydiagnostics.com>

22nd May 2025

Dear Dr. Yang,

Thank you for submitting your revised files. I am pleased to inform you that your manuscript is accepted for publication and is now being sent to our publisher to be included in the next available issue of EMBO Molecular Medicine.

Yours sincerely,

Lise Roth

Referee #3 (Comments on Novelty/Model System for Author):

N/A

Referee #3 (Remarks for Author):

I have no more questions to the authors. Congratulations for the work.
